# TUMTraf VideoQA: Dataset and Benchmark for Unified Spatio-Temporal Video Understanding in Traffic Scenes

Xingcheng Zhou [1]    Konstantinos Larintzakis [1]    Hao Guo [1]    Walter Zimmer [1]    Mingyu Liu [1]    Hu Cao [1]
Jiajie Zhang [1]    Venkatnarayanan Lakshminarasimhan [1]    Leah Strand [1]    Alois C. Knoll [1]

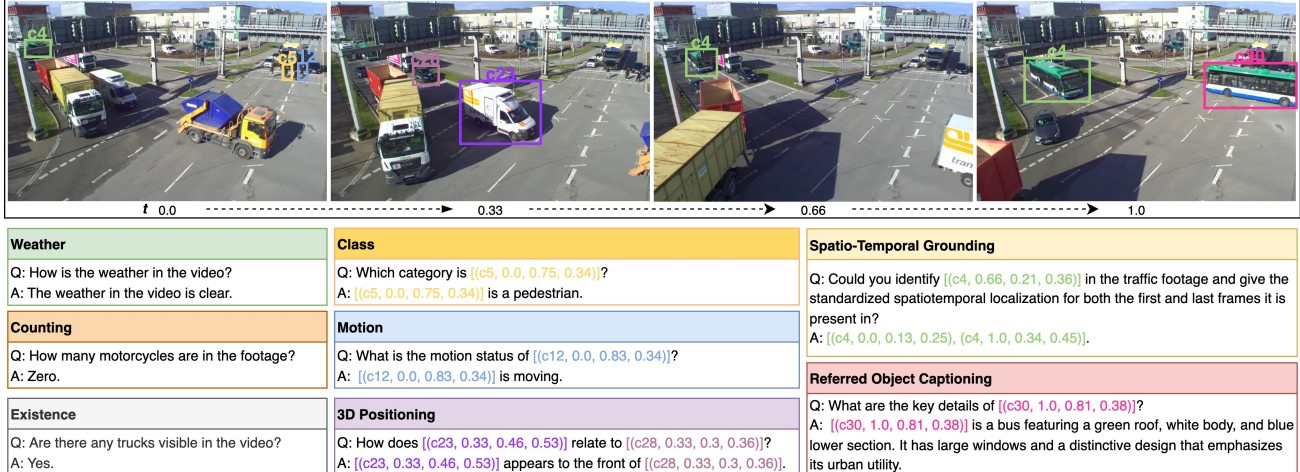

Figure 1: TUMTraf VideoQA introduces a comprehensive benchmark for video-level traffic scene understanding. Our baseline model, TraffiX-Qwen, is capable of solving multiple tasks, including video QA, spatio-temporal grounding, and referred object captioning, within a unified model. In our approach, the spatio-temporal location of objects is represented as tuples $(c, fn, x, y)$, where $c$ serves as a unique object identifier, $fn$ denotes the normalized frame timestamp, and $(x, y)$ denote the center of the object in the image, normalized with respect to the image dimensions.

## Abstract

We present TUMTraf VideoQA, a novel dataset and benchmark designed for spatio-temporal video understanding in complex roadside traffic scenarios. The dataset comprises 1,000 videos, featuring 85,000 multiple-choice QA pairs, 2,300 object captioning, and 5,700 object grounding annotations, encompassing diverse real-world conditions such as adverse weather and traffic anomalies. By incorporating tuple-based spatio-temporal object expressions, TUMTraf VideoQA unifies three essential tasks—multiple-choice video question answering, referred object captioning, and spatio-temporal object grounding—within a cohesive evaluation framework.

We further introduce the TraffiX-Qwen baseline model, enhanced with visual token sampling strategies, providing valuable insights into the challenges of fine-grained spatio-temporal reasoning. Extensive experiments demonstrate the dataset's complexity, highlight the limitations of existing models, and position TUMTraf VideoQA as a robust foundation for advancing research in intelligent transportation systems. The dataset and benchmark are publicly available to facilitate further exploration.

## 1. Introduction

With the advancement of intelligent roadside infrastructure and Large Language Models (LLMs) (Grattafiori et al., 2024), leveraging language to achieve a more generalized and interpretable understanding of traffic scenes becomes increasingly important. This involves accurately capturing the relationships among traffic participants, generating descriptive captions of their appearances, and analyzing their spatio-temporal positions and interactions (Zhang et al.,

[1]School of Computation, Information and Technology, Technical University of Munich, Germany. Correspondence to: Xingcheng Zhou <xingcheng.zhou@tum.de>, Hu Cao <hu.cao@tum.de>.

*Proceedings of the 42nd International Conference on Machine Learning*, Vancouver, Canada. PMLR 267, 2025. Copyright 2025 by the author(s).

2024; Zhou & Knoll, 2024). Traditional models for traffic scene understanding are typically designed for specific tasks, such as object recognition, object association, and traffic flow analysis. Although these methods have achieved notable success within isolated domains, they often face significant challenges in scalability, generalization to diverse traffic conditions, and real-world deployment. The emergence and rapid development of large foundation models (Liu et al., 2023; Zhou et al., 2024a) present new opportunities to address these challenges. These models offer the potential to overcome traditional limitations by leveraging their ability to generalize across multiple tasks, integrate multimodal information, and adapt to complex, dynamic traffic scenarios in a more flexible and unified manner.

Previous studies have primarily advanced traffic scene understanding through image-based question-answering tasks in driving environments (Sima et al., 2024; Zhou et al., 2024b; Qian et al., 2024b). However, image-level Vision-Language Models (VLMs) are inherently limited in their ability to capture the temporal dynamics crucial for comprehending complex traffic events. In contrast, intricate traffic scenarios often require multi-frame video analysis for accurate real-world understanding. Besides, despite the growing number of vision-language datasets developed for driving scenarios, a significant gap persists in the exploration of multimodal datasets specifically designed for the roadside traffic domain. In particular, video-based datasets captured from a third-party perspective and tailored to traffic scene understanding remain notably underexplored.

To bridge the gap in this domain, we propose TUMTraf VideoQA, a video language dataset designed to benchmark the model understanding capabilities in roadside traffic scenarios. The dataset encompasses video question-answering, object captioning, and spatio-temporal grounding tasks, capturing key elements crucial for understanding real-world traffic scenes. An illustrative example from the dataset is shown in Figure 1. The main contributions of this work can be outlined as follows:

- We present TUMTraf VideoQA, a comprehensive video-language dataset designed for complex traffic video understanding. The dataset captures a diverse range of real-world scenarios, including extreme weather conditions and critical corner cases such as traffic accidents.

- We propose a novel benchmark that evaluates model performance across three key tasks, including video question answering, referred object captioning, and spatio-temporal grounding, facilitating fine-grained reasoning in traffic scenarios.

- We establish the TraffiX-Qwen baseline and provide detailed results and analyses. Through extensive ex-

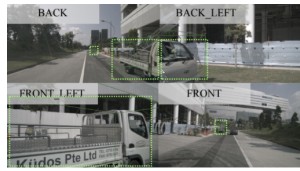

(a) Objects with the prompt: *A white truck that is stationary in the same direction.* (Wu et al., 2023b)

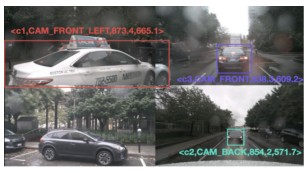

(b) Frame-based object expression using numerical coordinates (Sima et al., 2024).

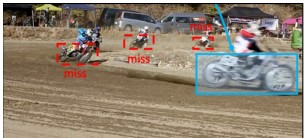

(c) Object referring in (Zhang et al., 2020) with prompt: *What is beneath the adult.*

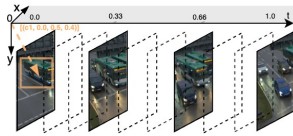

(d) Location of the green bus [(c1,0.0,0.5,0.4)] in the video. (Ours)

Figure 2: Different methods for describing objects in images and videos using language expressions. We adopt a tuple-based spatio-temporal object representation for the unique object reference, as shown in (d).

periments with various efficient visual token sampling strategies, we offer valuable insights and outline potential future research directions.

## 2. Related Work

### 2.1. Vision-Language Datasets in Traffic Scenes

A growing number of open-source datasets have been released to facilitate autonomous driving and intelligent transportation systems, such as BDD100k (Yu et al., 2020), Waymo Open Dataset (Sun et al., 2020), and Ego4D (Grauman et al., 2022). In recent years, the rapid advancements in LLMs drive significant efforts to integrate language for the development of vision-language foundation models in this domain. As summarized in Table 1, several pioneering datasets have been introduced for traffic scenarios, particularly focusing on vehicle-centric environments (Liu et al., 2024b). NuScenes-QA (Qian et al., 2024b) provides a question-answering benchmark tailored for driving scenes. Meanwhile, DRAMA (Malla et al., 2023) is designed for video-level open-ended tasks aimed at evaluating driving instructions and assessing the importance of objects within their environments. Besides, referring to specific traffic participants through natural language—commonly known as referred object grounding and tracking—is a crucial task in traffic scene understanding. Some works (Wu et al., 2023a;b) extend the KITTI (Geiger et al., 2013) and nuScenes (Caesar et al., 2020) datasets, by associating natural language descriptions with specific vehicles and pedestrians. This facilitates fine-grained identification and tracking of traffic participants, allowing for precise object localiza-

Table 1: Summary and comparison of visual-language datasets in the traffic domain for question answering, video grounding, and referred multi-object tracking. The table's upper section presents QA tasks, while the lower section covers grounding and referring tasks. We introduce the first roadside video understanding dataset and unify these tasks in one benchmark.

| Dataset | Venue | Tasks | QA Gen. | # Videos/Scenes | # QAs/Captions | # Grounding | Domain |
|---|---|---|---|---|---|---|---|
| DRAMA (Malla et al., 2023) | WACV'23 | Video QA | Manual | 18k | 102k | - | Driving |
| LingoQA (Marcu et al., 2024) | ECCV'24 | Video QA | Manual | 28k | 419k | - | Driving |
| NuScenes-QA (Qian et al., 2024b) | AAAI'24 | Image QA | Template | 850 | 460k | - | Driving |
| DriveLM (Sima et al., 2024) | ECCV'24 | Image QA | Temp. + Man. | 188k | 4.2M | - | Driving |
| City-3DQA (Sun et al., 2024) | ACM MM'24 | Scene QA | Temp. + Man. | 193 | 450k | - | City |
| HC-STVG (Tang et al., 2022) | ACM MM'22 | Video Grounding | Manual | 5.6k | - | 5.6k | General |
| DVD-ST (Ji et al., 2024) | - | Video Grounding | Manual | 2.7k | - | 5.7k | General |
| UCA (Yuan et al., 2024) | CVPR'24 | Video Ground. & Cap. | Manual | 1.8k | - | 23.5k | Surveillance |
| Refer-KITTI (Wu et al., 2023a) | CVPR'23 | Referred-MOT | Manual | 18 | - | 818 | Driving |
| NuPrompt (Wu et al., 2023b) | AAAI'25 | Referred-MOT | LLM | 850 | - | 35k | Driving |
| **TUMTraf VideoQA (Ours)** | ICML'25 | Video QA, ST Grounding | Temp. + LLM | 1k | 87.3k | 5.7k | Roadside |

tion based on language descriptions in complex driving environments. In the traffic domain, OATS (Agarwal & Chen, 2023) introduces a structured representation based on ordered atomic activities for fine-grained scenario understanding. Action-slot (Kung et al., 2024) further leverages slot attention to recognize multiple activities, achieving strong results on both synthetic and real-world datasets. However, most existing efforts primarily focus on driving scenarios and are typically constrained to individual tasks such as question answering, video grounding, or referred multi-object tracking. A significant research gap also remains in the availability of large-scale datasets designed specifically for roadside surveillance scenarios. Our work aims to bridge this gap by providing a comprehensive dataset tailored for multiple tasks in roadside traffic understanding within a unified framework.

## 2.2. Fine-Grained Video Understanding

Fine-grained video understanding centers on the precise analysis of intricate video content, targeting tasks that demand nuanced reasoning across spatial and temporal dimensions. Some representative tasks include spatio-temporal grounding (Zhang et al., 2020; Tang et al., 2022), mapping specific objects or events to precise locations and times within a video based on a given query; video object referring (Ding et al., 2023; Wu et al., 2023a;b), which involves tracking objects through space and time given text prompts; video temporal grounding (Lin et al., 2023; Huang et al., 2024), identifying specific moments or intervals in a video that align with a provided textual query. These tasks require high-precised, nuanced multimodal alignment, and the ability to capture subtle temporal and spatial dynamics, making it particularly challenging. With the progress in visual LLMs, recent works enhance capabilities of video understanding (Tang et al., 2023) to facilitate comprehension across both abstract and fine-grained levels, some approaches introduced enhanced video representations via self-supervised learning (Qian et al., 2024a), while several

methods focused on improving cross-modality tuning and alignment (Bi et al., 2025a; Gao et al., 2023; Bi et al., 2025b). Despite these advances, achieving structured and precise fine-grained feature representations in video understanding remains an open challenge.

## 2.3. Language-Based Object Referring

Referring objects in visual data, such as images and videos, is typically achieved by associating them with predefined definitions or language descriptions. Figure 2 illustrates four commonly used methods for representing objects through language expressions. The inherent ambiguity of natural language, coupled with the modality gap between visual and linguistic representations, presents significant challenges. Object representation in tasks such as object referring often necessitates careful dataset curation to ensure that linguistic expressions uniquely or collectively correspond to specific objects in videos. For example, some datasets include only scenarios with uniquely identifiable objects (Tang et al., 2022), while others contain expressions that jointly refer to multiple objects (Ji et al., 2024). However, in complex real-world applications such as autonomous driving, textual descriptions alone are often insufficient to uniquely specify an object. To address this challenge, DriveLM (Sima et al., 2024) introduces a structured tuple representation, $<c, CAM, x, y>$, where c denotes the object identifier, CAM specifies the camera, and $<x, y>$ represents the 2D center coordinates within the camera's coordinate system. Alternatively, ELM (Zhou et al., 2024b) simplifies the problem by converting temporal video tasks into frame-level questions, using a tuple $<c, x, y>$ to identify objects within individual frames without temporal dependencies. Despite the advancements, formulating a unified, precise, and unique language representation for objects in video remains an open challenge.

In this work, we design a spatio-temporal object representation in videos with a four-element tuple format $(c, f_n, x, y)$,

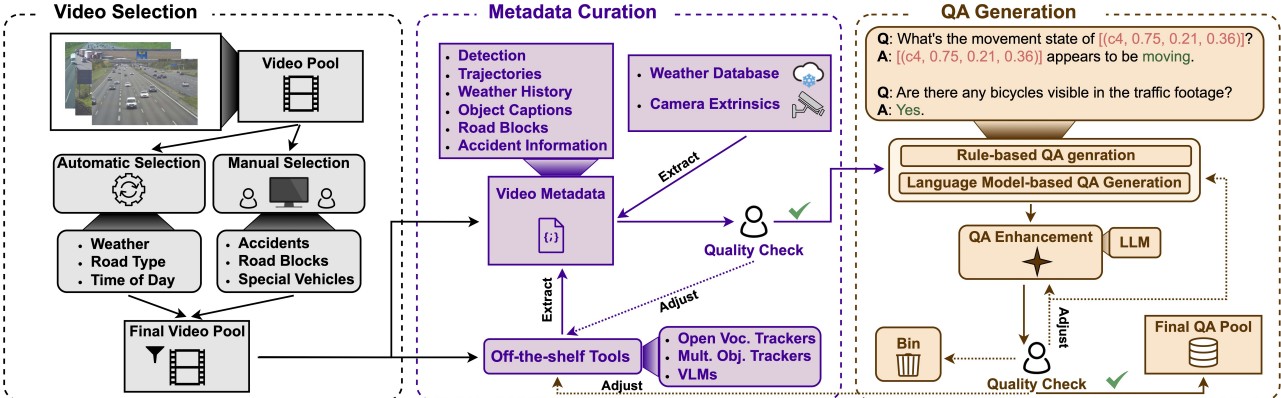

Figure 3: The workflow of the semi-automatic annotation pipeline for TUMTraf VideoQA generation, integrating external database, leveraging various off-the-shelf tools and LLMs, with human quality checks ensuring accuracy.

where c denotes a unique object identifier, $f_n$ indicates the normalized frame timestamp, and $(x, y)$ corresponds to the object's normalized spatial coordinates within the frame. The same object is consistently assigned the identifier c throughout the video, while its spatial position changes over time. This formulation enables precise tracking and referencing of objects across both spatial and temporal dimensions, facilitating robust language-based interaction in dynamic environments. Besides, it provides a standardized interface for fine-grained video understanding, enabling more detailed and structured analysis.

## 3. TUMTraf VideoQA Dataset

### 3.1. Dataset Creation

Our data generation process comprises three primary stages: Video Selection, Metadata Curation, and QA Pair Generation, as shown in Figure 3. To ensure high-quality, diverse, and balanced annotations, we introduce a semi-automatic labeling pipeline that combines automated processes with human verification for enhanced accuracy and consistency.

**Video Selection.** The video data in TUMTraf VideoQA are collected from multiple roadside infrastructure points over a data collection period spanning more than two years. The dataset encompasses diverse perspectives, covering various urban, suburban, and highway scenarios. It includes a broad range of video content, capturing various distinct traffic scenarios, such as traffic accidents, rescue operations, congestion, roadblocks, and uncommon vehicle occurrences. Furthermore, the dataset encompasses a variety of environmental conditions, including sunny, rainy, cloudy, snowy, and foggy weather, along with technical challenges scenarios such as obstructed camera lenses and vibrations. The video segments are carefully selected to include a diverse range of traffic participants—including vehicles, pedestrians, and obstacles—capturing the complexity and dynamic

characteristics of real-world traffic environments.

**Metadata Curation.** The video metadata includes environmental conditions, object positions, trajectories, appearances, traffic flows, and more, serving as the basis for generating high-quality annotations. External data sources include historical weather records, traffic accident reports, and camera calibration details. To ensure precise time-specific weather and traffic information, we align video timestamps with these records using GPT-4o and Text-embedding-3-large (OpenAI et al., 2024). For visual metadata, we utilize state-of-the-art object detectors and trackers (Wang et al., 2024; Zhao et al., 2024), along with open-vocabulary detectors (Yan et al., 2023; Wu et al., 2024), to generate bounding box and trajectory data. We then transform 2D information into camera-based pseudo-3D locations using camera calibration matrices, facilitating the generation of questions related to object motion and relative spatial positioning. To capture object appearance details, we utilize large VLMs (OpenAI et al., 2024; Liu et al., 2024a), which automatically generate textual descriptions for cropped object bounding boxes. A manual quality assurance step is conducted to thoroughly evaluate the accuracy and completeness of the metadata. Any identified deficiencies trigger necessary adjustments and a reprocessing cycle to ensure data quality and integrity before progressing to the next stage.

**QA Generation & Filtering.** To ensure a balance between question diversity and accuracy, we adopt a hybrid approach that combines template-based and LLM-driven generation strategies. Approximately 15 question templates are manually designed for each question type and further expanded using LLMs-generated variations. These templates are populated with relevant objects and metadata to generate initial QA pairs using GPT-4o-mini. The LLM is then prompted to refine the generated content by rephrasing either the question alone or both the question and its corresponding answer, depending on the context. Once QA pairs are generated for

each question type, a selective quality evaluation is conducted to assess their accuracy and relevance. This iterative process involves refining question templates, adjusting off-the-shelf tools, and discarding QA pairs that do not meet the predefined quality standards. The validated QA pairs are then integrated into the TUMTraf VideoQA dataset, ensuring high-quality and diverse annotations.

### 3.2. Tasks and Metrics

TUMTraf VideoQA benchmark comprises three core tasks to thoroughly evaluate model performance in traffic scenes: Multi-Choice Question Answering (MQA), Video Referred Object Captioning (V-ROC), and Spatio-Temporal Object Grounding (ST-OG). QA pairs related to weather and traffic accidents are included for training and future research but are not considered in the benchmark evaluation.

**Multi-Choice Question Answering.** The MQA task assesses the model's capabilities across five key dimensions: **Positioning**, identifying the relative 3D spatial location of objects; **Counting**, determining the number of occurrences of a particular object or class across the video; **Motion**, analyzing the movement status of objects; **Class**, categorizing objects based on their type or attributes; **Existence**, querying whether a specific object or category is present in the video. Following (Qian et al., 2024b), each dimension is further divided into easy and hard levels, depending on whether the question requires single-hop or multi-hop reasoning. We show the template of easy and hard questions in Sec C.2. We use Top-1 accuracy as the evaluation metric and report the mean accuracy across all question types.

**Video Referred Object Captioning.** The task evaluates the model's capability to describe the appearance of a specified object in natural language. It aims to generate detailed and accurate summaries that effectively capture the object's key visual attributes. Unlike the image-based referred object captioning task (Sima et al., 2024; Zhou et al., 2024b), we query an object based on its spatial and temporal location within a video, which adds a significant level of complexity. In this task, we adopt common NLG metrics (Sai et al., 2022), including BLEU, CIDEr, ROUGE, METEOR, and SPICE, to measure the quality of descriptions.

**Spatio-Temporal Object Grounding.** Accurately identifying the spatio-temporal positions of a specified object is crucial in traffic scenarios. Unlike traditional video grounding (Tang et al., 2022) or referred multi-object tracking tasks (Wu et al., 2023b), which primarily focus on locating objects within individual frames across the video, ST-OG simplifies the process by providing start and end frames along with corresponding spatial coordinates in a standardized tuple format: $[(c, f_n', x', y'), (c, f_n'', x'', y'')]$. This task serves to assess a model's performance in effectively associating objects across frames while accurately determining their

temporal extent and spatial positions within the video.

We adopt three evaluation metrics to assess the performance of this task, i.e., Temporal error $\mathcal{E}_{f_n}$, Spatial error $\mathcal{E}_s$ and Spatio-Temporal error $\mathcal{E}_{st}$. Temporal error $\mathcal{E}_{f_n}$ and Spatial error $\mathcal{E}_s$ use the L1 loss, which measures the absolute temporal differences $\Delta f_n$ and the spatial displacement $\Delta s = \|(\Delta x, \Delta y)\|_2$. The Spatio-Temporal error $\mathcal{E}_{st}$ adopts L2 loss and captures deviations across both spatial and temporal dimensions. For each metric, both the start and end frames are considered, with the formulations as follows:

$$\mathcal{E}_{f_n} = \frac{\Delta f_n' + \Delta f_n''}{2}; \quad \mathcal{E}_s = \frac{\Delta s' + \Delta s''}{2} \tag{1}$$

$$\mathcal{E}_{st} = \frac{1}{2}\left(\|(\Delta f_n', \Delta x', \Delta y')\|_2 + \|(\Delta f_n'', \Delta x'', \Delta y'')\|_2\right) \tag{2}$$

### 3.3. Dataset Statistics

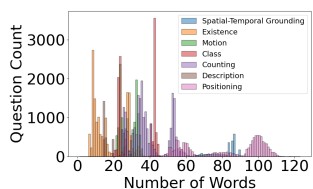

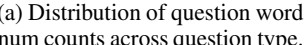

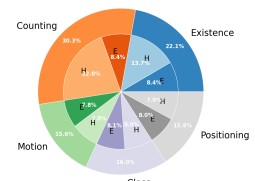

(a) Distribution of question word num counts across question type.

(b) Class distribution of Multi-Choice QA.

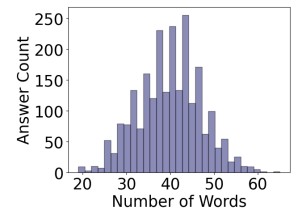

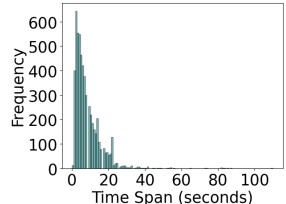

(c) Distribution of answer word counts in Video Referred Object Captioning.

(d) Temporal window lengths in Spatio-Temporal Grounding.

Figure 4: Statistical distributions of the TUMTraf VideoQA dataset, including word counts in questions and answers, distribution of question types, and temporal window lengths for object grounding.

TUMTraf VideoQA dataset consists of 1,000 videos, 85,000 multi-choice QA pairs, 5,700 spatio-temporal grounding prompts, and 2,300 referred object captioning. Video durations range from 10 seconds to 2 minutes. We split the videos into training and validation sets with a ratio of 7:3, ensuring that videos in the validation set do not overlap with those in the training set. Generated QA pairs inherit the split of their associated videos, forming distinct videos and annotations for training and validation. Figure 4 provides an overview of the dataset's statistical distributions, including question complexity, question-type distribution, answer lengths, and the temporal window distribution of queried

objects in the spatio-temporal grounding task. Figure 5 shows video statistics of the TUMTraf VideoQA dataset, highlighting its diversity across scene types, traffic conditions, and object categories. We categorize scenes into three types: highways (rural), urban intersections (city), and country roads (rural/urban). The traffic volume trends, track durations, and category distributions across different times and locations collectively indicate that the dataset captures realistic traffic dynamics and reflects patterns consistent with real-world distributions. These statistics underscore the broad coverage and its suitability for training and evaluating fine-grained video understanding models in diverse traffic scenarios. Further details and statistics are available in Sec. A.1.

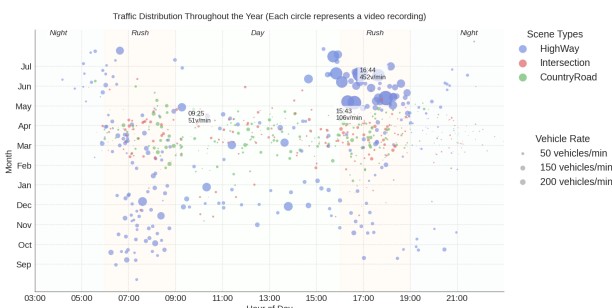

(a) Traffic distribution throughout the year by time of day and scene type.

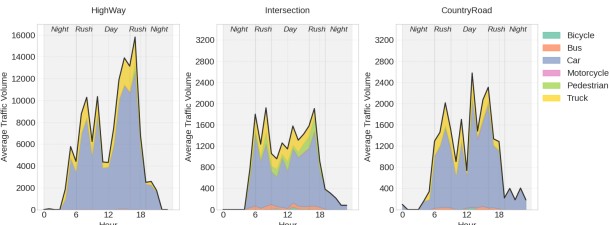

(b) Track duration of different road users across scene types.

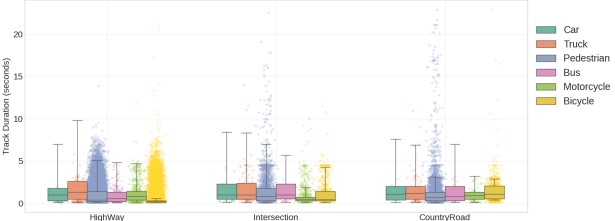

(c) Average hourly traffic volume by scene type and object class.

Figure 5: TUMTraf VideoQA dataset video distribution over scene types, traffic conditions, and categories.

## 4. TraffiX-Qwen Baseline

### 4.1. Model Architecture

We introduce TraffiX-Qwen, a baseline model for the TUMTraf VideoQA dataset that effectively addresses all three

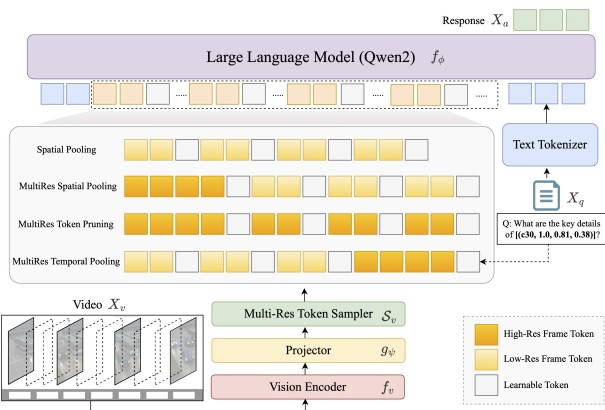

Figure 6: Overview of the TraffiX-Qwen baseline model. Yellow and orange colors represent the combination of multi-resolution visual tokens from different visual strategies, while blue indicates textual tokens.

tasks within a unified framework. The architecture of the TUMTraf VideoQA baseline, as illustrated in Figure 6, consists of four core components: visual encoder $f_v$, cross-modality projector $g_\psi$, token sampler $\mathcal{S}_v$, and large language model $f_\phi$, following (Li et al., 2024).

**Visual Encoder.** The video is uniformly divided into 100 segments, including the first and last frames, resulting in a total of $N = 101$ frames. Given the sampled video input $\mathbf{X} \in \mathbb{R}^{N \times H \times W \times 3}$, we adopt SigLIP (Zhai et al., 2023), a Transformer-based model pre-trained on large-scale language-image datasets, as the visual encoder. Each frame is processed at a resolution of $384 \times 384$, and the video is encoded into a sequence of visual features $Z_v = [v_1, \ldots, v_N]$, where $v_i = f_v(\mathbf{X}_i) \in \mathbb{R}^{T \times C}$, containing $T$ spatial tokens of dimension $C$.

**Token Sampling Strategy.** We leverage a simple yet effective frame-level multi-resolution sampling strategy to enhance feature representation. We evaluate four primary sampling strategies: spatial pooling, multi-resolution spatial pooling, multi-resolution token pruning, and multi-resolution temporal pooling. The output $Z_v$ from the last layer of SigLIP is denoted as $Z_{high}$, which is reduced to $T'$ tokens after down-sampling. We define the set of high-resolution frames as keyframes, denoted by $\mathcal{K}(\cdot)$. Additionally, a learnable token is appended to the end of each frame to explicitly differentiate them. The number of tokens used in various strategies is presented in Table 2.

• **Spatial Pooling**: This method applies spatial pooling to each feature map $Z_{high}$, resulting in a down-sampled representation $Z_{low} = f_{pool}(Z_{high})$ with $N \times T'$ tokens, as shown in Eq. 3. We use the notation $[\cdot]_n^N$ to represent the operation of sequentially concatenating the processed feature maps.

Table 2: Comparison of visual token numbers across different token sampling strategies. We keep the high resolution at 27×27 and the low resolution at 14×14.

| Method | Number of Visual Tokens | Max Tokens |
|---|---|---|
| Spatial Pooling | $N \times T' + N$ | 19,897 |
| MultiRes Spatial-Pooling | $T + (N-1) \times T' + N$ | 20,430 |
| MultiRes Token-Pruning | $T + (N-1) \times r \times T + N$ | 18,574 |
| MultiRes Temporal-Pooling | $K \times T + (N-K) \times T' + N$ | 20,963 |

$$S_v(Z_v) = [Z_{\text{low}}^n, Z_{\text{learn}}]_{n=1}^N \quad (3)$$

• **MultiRes Spatial Pooling**: Compared to the naive spatial pooling, this strategy selects the first frame as the keyframe $\mathcal{K} = (1)$, and is retained at its original resolution $Z_{\text{high}}^1$. It is formulated in Eq. 4.

$$S_v(Z_v) = [Z_{\text{high}}^1, Z_{\text{learn}}, [Z_{\text{low}}^n, Z_{\text{learn}}]_{n=2}^N] \quad (4)$$

• **MultiRes Token Pruning**: Similar to MultiRes Spatial Pooling, the first frame is designated as the keyframe. Token-wise cosine similarity is then computed between the keyframe and each subsequent frame, while visual tokens with lowest similarity are selectively retained based on predefined ratio $r$, formulated as $Z_{\text{pruned}} = f_{\text{prune}}^r(Z_{\text{high}})$, shown in Eq. 5. To ensure visual token efficiency comparable to spatial pooling, $r$ is set to 0.25. A similar strategy is also applied in autonomous driving scenarios (Ma et al., 2024).

$$S_v(Z_v) = [Z_{\text{high}}^1, Z_{\text{learn}}, [Z_{\text{pruned}}^n, Z_{\text{learn}}]_{n=2}^N] \quad (5)$$

• **MultiRes Temporal Pooling**: In this strategy, the keyframe set is adaptively queried by input questions $\mathcal{K}(\cdot) = \mathcal{Q}(X_q)$. Based on the temporal regions of interest derived from the question, $K$ keyframes are selected, which are preserved with high-resolution representations $Z_{\text{high}}^n$. Meanwhile, the remaining frames undergo spatial pooling, resulting in $Z_{\text{low}}^n$, as expressed in Eq. 6. Typically, $K \leq 2$, and for general questions without specific temporal focus, the first frame is set as the default keyframe.

$$S_v(Z_v) = [Z_v^n, Z_{\text{learn}}]_{n=1}^N$$
$$\text{where } Z_v^n = \begin{cases} Z_{\text{high}}^n, & \text{if } n \in \mathcal{K}(\cdot), \\ Z_{\text{low}}^n, & \text{if } n \notin \mathcal{K}(\cdot) \end{cases} \quad (6)$$

**Large Language Model.** We adopt Qwen-2 (Yang et al., 2024) as the pre-trained LLM in our TraffiX-Qwen baseline. Qwen-2 demonstrates strong capabilities in in-context learning and instruction following, supporting context lengths of up to 32k tokens. This allows for the processing of complex and long-form inputs effectively. We utilize two versions

of Qwen-2, namely 0.5B and 7B, to establish baselines of different scales. The answer generation process in our TraffiX-Qwen baseline model is formulated as:

$$p(X_a \mid S_v(Z_v), X_q) = \prod_{t=1}^{\mathcal{T}} P_{\phi,\psi}(x_t \mid x_{1:t-1}, S_v(Z_v), X_q) \quad (7)$$

### 4.2. Baseline Training

Our baseline model undergoes a two-stage training process consisting of video-language alignment and visual instruction fine-tuning, to enhance its understanding of traffic scenarios and reasoning capabilities for long videos. Both stages are trained with 4 NVIDIA A100 GPUs.

**Video-Language Alignment.** This step aims to align video representations with language embeddings, ensuring that the LLM can effectively interpret the visual features. We freeze both the visual encoder and the LLM, and train only the projector layer. To facilitate the training, we initialize the parameters of the 2-layer MLP from the LLaVA-OneVision model, which has been pre-aligned with large-scale cross-modality datasets, including 3.2M single-image and 1.6M OneVision image-caption pairs. In this stage, we further train the projector on raw TUMTraf VideoQA data, utilizing open-ended captioning pairs without transforming it into the multiple-choice QA for 1 epoch.

**Visual Instruction Fine-Tuning.** Building upon the robust representations established during the alignment stage, we further fine-tune our baseline model on the training set of TUMTraf VideoQA. The multi-choice QA pairs are reformatted into the instruction-following format to prompt the model to generate the corresponding answers. During this stage, we freeze the vision encoder and projector layers and finetune the Qwen-2 model with full-parameter fine-tuning to adapt its reasoning and contextual understanding to the traffic environment. The model is fine-tuned for 1 epoch.

Table 4: Training and inference cost of TraffiX-Qwen.

| Ver. | #Vision | #Proj. | #LLM | Inf./QA | #Trainable | Train Hour |
|---|---|---|---|---|---|---|
| 0.5B | 397.8M | 1.8M | 493.8M | ~1.6s | 495.6M | 28h |
| 7B | 397.8M | 17.0M | 7612.6M | ~3.8s | 7629.6M | 36h |

**Computational Cost.** Table 4 summarizes the training and inference details of the TraffiX-Qwen models. We show the number of parameters of the vision backbone, the projector, and the LLMs. Inference speed is reported as the average time per question, using autoregressive decoding on a single A100 GPU, without acceleration strategies such as quantization or caching.

Table 3: Evaluation of Open-source models and TraffiX-Qwen baseline on the Multi-Choice QA track of the TUMTraf VideoQA Dataset, where **E** represents easy, single-hop questions, and **H** denotes hard, multi-hop questions.

| Models | Category | Positioning | | Counting | | Motion | | Class | | Existence | | Overall |
|---|---|---|---|---|---|---|---|---|---|---|---|---|
| | | E | H | E | H | E | H | E | H | E | H | |
| Open-Source Models | | | | | | | | | | | | |
| LLAVA-OneVision (Li et al., 2024) | 0.5B | 42.10 | 25.26 | 27.62 | 30.45 | 54.87 | 37.04 | **57.06** | 39.57 | **85.29** | 58.35 | 45.82 |
| | 7B | **46.92** | 22.03 | **69.42** | 54.85 | 61.14 | 60.48 | 51.92 | **56.50** | 77.08 | 63.25 | **56.36** |
| Qwen2-VL (Bai et al., 2023) | 2B | 36.73 | **26.05** | 38.10 | 39.78 | 56.46 | 35.19 | 32.10 | 38.49 | 68.87 | 67.32 | 43.91 |
| | 7B | 36.03 | 24.35 | 66.91 | 49.11 | **61.65** | 38.10 | 44.83 | 40.20 | 54.00 | **73.03** | 48.82 |
| VideoLLaMA2 (Cheng et al., 2024) | 2.0-7B-8F | 42.54 | 18.14 | 44.13 | 37.56 | 59.37 | 35.87 | 39.05 | 44.07 | 44.56 | 65.56 | 43.09 |
| | 2.0-7B-16F | 42.41 | 10.47 | 55.98 | 41.94 | 53.80 | 52.26 | 44.16 | 47.75 | 66.93 | 64.82 | 48.05 |
| TraffiX-Qwen Baseline | | | | | | | | | | | | |
| Baseline-0.5B (Ours) | Spatial Pooling | 75.54 | 68.47 | 85.31 | 75.82 | 83.92 | **81.26** | 79.95 | 59.73 | 93.06 | 85.37 | 78.84 |
| | MultiRes Spatial-Pooling | 76.36 | 69.32 | 86.10 | 75.86 | 83.73 | 79.59 | **80.57** | 61.70 | 92.73 | 85.37 | 79.07 |
| | MultiRes Token-Pruning | **76.61** | 73.40 | **86.33** | 76.88 | 83.48 | 78.60 | 80.01 | 60.43 | **93.34** | 85.27 | 79.44 |
| | MultiRes Temporal-Pooling | 75.85 | **74.07** | 85.65 | 76.92 | 84.05 | 80.64 | 80.26 | **62.21** | 93.06 | **85.55** | 79.83 |
| Baseline-7B (Ours) | Spatial Pooling | 76.99 | 76.14 | 87.07 | 76.81 | 86.58 | 82.07 | 82.72 | 64.11 | 93.62 | 85.27 | 81.14 |
| | MultiRes Spatial-Pooling | **78.89** | 76.99 | 87.07 | 77.49 | **88.29** | 81.82 | **83.52** | 65.95 | 93.01 | **85.51** | 81.85 |
| | MultiRes Token-Pruning | 76.93 | 77.24 | 87.41 | 77.76 | 86.46 | 80.64 | 82.66 | 65.00 | **93.84** | 85.48 | 81.34 |
| | MultiRes Temporal-Pooling | 78.57 | 77.24 | **87.53** | 78.22 | 87.09 | **82.68** | 83.33 | 65.76 | 93.78 | 85.34 | **81.95** |

## 5. Experiments

Extensive experiments are conducted on the TUMTraf VideoQA dataset. We evaluate SOTA open-source VLMs in a zero-shot setting to assess their spatio-temporal reasoning abilities, analyze the dataset's characteristics, and examine the impact of different visual sampling strategies on performance. During inference, the temperature is set to zero to ensure deterministic outputs and enhance consistency.

### 5.1. Quantitative Results in Multi-Choice QA

Table 3 presents the quantitative results in this task, offering several key insights, which are summarized as follows.

**Difficulty of Question Types.** The accuracy across different question types reveals consistent trends of difficulty for both open-source VLMs and our baseline models. Among the evaluated question types, existence questions are the least challenging, achieving the highest accuracy. This is followed by counting and motion questions, which necessitate the extraction and reasoning of information across multiple video frames. In contrast, positioning questions, which require a deeper understanding of 3D spatial relationships, emerge as the most challenging. Moreover, the accuracy of multi-hop questions is generally lower compared to single-hop questions, reflecting the increased complexity of complex reasoning tasks that demand the capture of more fine-grained details and intricate reasoning processes.

**Open-Source Model Performance.** We evaluate the performance of three open-source models: LLaVA-OneVision (Li et al., 2024), Qwen2-VL (Bai et al., 2023), and VideoLLaMA2 (Cheng et al., 2024) on our Multi-Choice QA task. The results indicate that increasing model size significantly enhances their performance in zero-shot video QA scenarios, with improvements from 5% to 10%. Notably, VideoLLaMA2 benefits from incorporating more frames, leading to a notable boost in accuracy. Among the three models with 7B parameters, Qwen2-VL and VideoLLaMA2 achieve comparable overall performance, whereas LLaVA-OneVision outperforms both, achieving the highest accuracy. Furthermore, all models struggle with positioning questions, highlighting their limitations in spatial reasoning.

**Effect of Token Sampling Strategy.** Experimental results from the 0.5B and 7B baseline models demonstrate that multi-resolution strategies can enhance model performance to some extent, with MultiRes Temporal Pooling yielding the most significant gains. Notably, the MultiRes strategy can greatly improve positioning tasks that rely on spatial recognition, while having minimal impact on existence and counting tasks. Moreover, MultiRes Token Pruning effectively enhances positioning and counting accuracy but may inadvertently discard critical visual tokens, leading to limited or adverse effects on motion and existence tasks. While MultiRes Temporal Pooling enhances fine-grained reasoning, it has little impact on easy recognition tasks like existence. Although multi-resolution methods provide richer multi-granularity visual representations, the overall performance improvements remain moderate.

### 5.2. Results in Spatio-Temporal Grounding

The quantitative results for the Spatio-Temporal Grounding task, presented in Table 5, underscore the complexity of the

Table 5: Evaluation of Spatio-Temporal Errors Across Open-Source models and TraffiX-Qwen Baseline.

| Model | Temporal E↓ | Spatial E↓ | ST E↓ |
|-------|------------|-----------|-------|
| Open-Source Models | | | |
| LLAVA-OneVision (0.5B) | 0.7285 | 0.7212 | 0.8415 |
| LLAVA-OneVision (7B) | 0.7680 | 0.7750 | 0.8142 |
| Qwen2-VL (2B) | 0.7729 | 0.7793 | 0.8127 |
| Qwen2-VL (7B) | 0.7615 | 0.7647 | 0.8032 |
| VideoLLaMA2 (7B-8F) | **0.6225** | **0.6360** | **0.6896** |
| VideoLLaMA2 (7B-16F) | 0.7218 | 0.7383 | 0.7895 |
| TraffiX-Qwen Baseline | | | |
| 0.5B-Spatial-Pooling | 0.1220 | **0.1892** | **0.2600** |
| 0.5B-MultiRes-Spatial-Pooling | **0.1211** | 0.1894 | 0.2607 |
| 0.5B-MultiRes-Token-Pruning | 0.1230 | 0.1934 | 0.2650 |
| 0.5B-MultiRes-Temporal-Pooling | 0.1228 | 0.1912 | 0.2629 |
| 7B-Spatial-Pooling | **0.1083** | **0.1737** | **0.2382** |
| 7B-MultiRes-Spatial-Pooling | 0.1136 | 0.1822 | 0.2493 |
| 7B-MultiRes-Token-Pruning | 0.1152 | 0.1748 | 0.2454 |
| 7B-MultiRes-Temporal-Pooling | 0.1166 | 0.1790 | 0.2496 |

Table 6: Performance of Open-Source models and TraffiX-Qwen on Referred Object Captioning.

| Model | Bleu_4 | ROUGE_L | CIDEr | METEOR | SPICE |
|-------|--------|---------|-------|--------|-------|
| Open-Source Models | | | | | |
| LLAVA-OneVision (0.5B) | 0.48 | 10.16 | 0.0102 | - | - |
| LLAVA-OneVision (7B) | 5.77 | 14.09 | 0.1326 | - | - |
| Qwen2-VL (2B) | 8.72 | 17.93 | 0.2086 | - | - |
| Qwen2-VL (7B) | **10.47** | **20.14** | **0.4119** | - | - |
| VideoLLaMA2 (7B-8F) | 6.25 | 19.94 | 0.2391 | - | - |
| VideoLLaMA2 (7B-16F) | 6.87 | 18.69 | 0.2111 | - | - |
| TraffiX-Qwen Baseline 0.5B | | | | | |
| Spatial-Pooling | 34.99 | 50.44 | 2.5195 | 35.24 | 46.35 |
| MultiRes Spatial-Pooling | 34.91 | 50.26 | 2.4306 | 35.20 | 45.75 |
| MultiRes Token-Pruning | 35.07 | 50.79 | **2.5730** | 35.30 | 46.48 |
| MultiRes Temporal-Pooling | **35.63** | **51.00** | 2.5464 | **35.77** | **47.17** |
| TraffiX-Qwen Baseline 7B | | | | | |
| Spatial-Pooling | 36.74 | 52.04 | 2.5613 | 36.42 | 47.32 |
| MultiRes Spatial-Pooling | 37.60 | **53.26** | 2.6113 | **37.31** | **49.16** |
| MultiRes Token-Pruning | **37.83** | 52.31 | **2.6162** | 36.56 | 47.80 |
| MultiRes Temporal-Pooling | 37.48 | 52.58 | 2.4236 | 36.85 | 48.53 |

task. Findings across temporal, spatial, and spatiotemporal errors exhibit a general consistency, revealing that without fine-tuning, open-source VLMs struggle to understand the task and cannot accurately regress the corresponding tuples, leading to unreliable temporal and spatial localization. For the fine-tuned TraffiX-Qwen baseline models, multi-resolution strategies appear to diminish spatial and temporal grounding performance, in contrast to their effectiveness in Multi-Choice QA and Referred Object Captioning tasks. This suggests that while multi-resolution techniques enhance frame-based object recognition by providing finer visual details, dynamically adjusting frame-level resolution can introduce ambiguity in inter-frame representations, adversely affecting temporal grounding and, consequently, spatial localization capabilities across the video.

### 5.3. Results in Referred Object Captioning

As shown in Table 6, Qwen2-VL (7B) surpasses all other open-source models by a considerable margin, demonstrating its strong performance on referred object captioning task. For baseline models, both the 0.5B and 7B variants exhibit performance improvements across various metrics when enhanced with multi-resolution strategies. Moreover, the 7B models consistently outperform their smaller counterparts in both open-source and fine-tuned baseline settings. The impact of the visual token sampling strategy, however, varies with model size. MultiRes Temporal Pooling yields the most significant gains for the 0.5B model, whereas MultiRes Spatial Pooling proves most effective for the 7B models.

## 6. Conclusions and Future Works

In this work, we introduce TUMTraf VideoQA, a novel benchmark that aims at advancing spatio-temporal video understanding in complex real-world traffic scenarios. The dataset provides a large-scale collection of high-quality videos and annotations specifically curated for roadside surveillance, covering three fundamental tasks: multi-choice video QA, spatio-temporal grounding, and referred object captioning within a unified evaluation framework. Extensive evaluations using SOTA vision language models, along with the introduction of the TraffiX-Qwen baseline model, establish a strong foundation for future research and development. We believe further optimization, such as quantization, and pruning, is a promising direction for improving its deployment efficiency in traffic monitoring. TUMTraf VideoQA serves as a comprehensive benchmark to facilitate further advancements in traffic video analysis and contribute to the development of next-generation traffic foundation models.

## Impact Statement

This paper presents work whose goal is to advance the field of Machine Learning. There are many potential societal consequences of our work, none of which we feel must be specifically highlighted here.

## Acknowledgment

This research was supported by the project AUTOtech.agil (Grant No.01IS22088U) and the Federal Ministry of Education and Research of Germany (BMBF).
This work was also supported by the project VIDETEC-2 (Grant No.01F2232E) and the Federal Ministry for Digital and Transport of Germany (BMDV).

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

# A. TUMTraf VideoQA Dataset

## A.1. Dataset Statistics

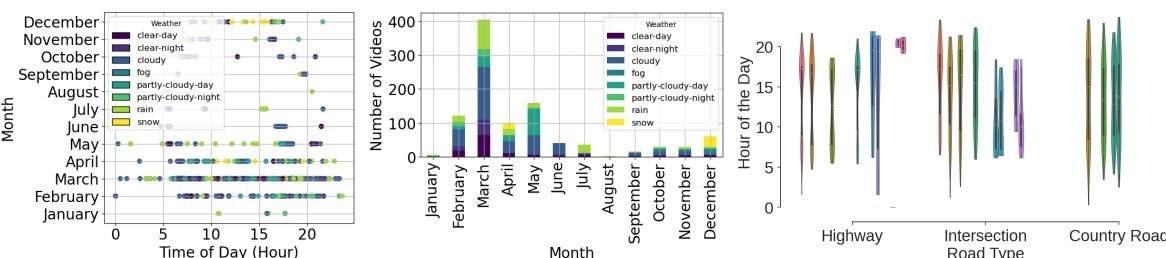

(a) Temporal Distribution of Video (b) Weather-Based Distribution of (c) Scene Distribution Across Different Perspec-
Weather Conditions Over the Years.    Videos.                                        tives.

Figure 7: Dataset distribution of video recordings by time, weather conditions, and perspectives.

The video selection process is meticulously designed to ensure comprehensive coverage of diverse daytime periods, weather conditions, road types, etc. The distribution of the video statistics in the TUMTraf VideoQA dataset is illustrated in Figure 7. Figure 7a provides an overview of the distribution of videos by hour of the day and month, with weather conditions represented through color coding. The majority of traffic footage was captured between 5:00 AM and 8:00 PM, with fewer recordings available during hours with limited natural light. Figure 7b illustrates the distribution of videos by weather conditions for each month. The dataset predominantly includes videos recorded between February and May, a period characterized by a wide variety of weather scenarios, thereby enhancing the dataset's representativeness. Figure 7c depicts the distribution of video recordings by hour of the day for each camera type and camera. The three camera categories—surveillance cameras positioned on highways, intersections, and country roads—are represented proportionately, ensuring video coverage across these categories from dawn to nighttime.

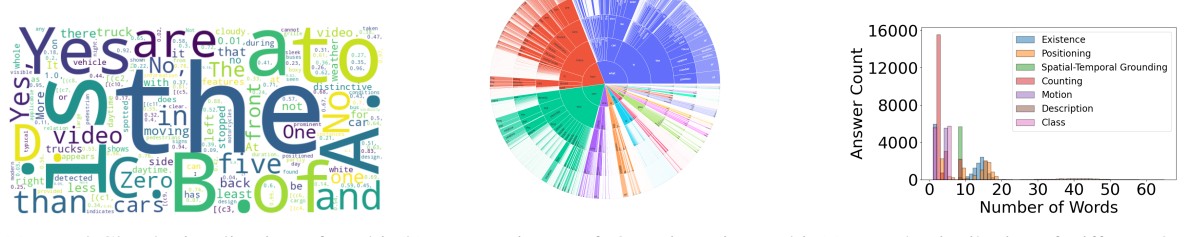

(a) Word Cloud Visualization of Multi-(b) Burst Figure of Questions in Multi-(c) Length Distribution of Different Ques-
Choice QA.                              Choice QA.                                 tion Types.

Figure 8: Distributions of video recordings across time, weather conditions, and camera types in the dataset.

In addition to video statistics, Figure 8 illustrates the distribution and characteristics of annotations in the TUMTraf VideoQA dataset. Figures 8a depict word clouds for answers across all three tasks, highlighting common terms and their frequencies. Figure 8b presents a sunburst chart that visualizes the distribution of question formats, revealing that most questions begin with "How," "What," and "Can". Figure 8c shows the distribution of answer lengths, indicating that the majority of answers consist of fewer than 10 words, with only a small number exceeding 19 words.

## A.2. Spatial Question Curation

Comprehending spatial relationships in 3D space is a critical challenge in traffic scene analysis. In our semi-automatic annotation pipeline, we calculate spatial locations by projecting 2D coordinates into 3D space under the planar assumption, leveraging historical camera intrinsic and extrinsic matrices. Specifically, from a third-party roadside perspective, we formulate spatial reasoning questions by treating each object as an ego-centric reference and formulate the questions that reveal its 3D positional relationships with surrounding traffic participants.

$$\text{relative position} = \begin{cases} \text{front} & \text{if } -15° < \theta \le 15° \\ \text{front left} & \text{if } 15° < \theta \le 75° \\ \text{left} & \text{if } 75° < \theta \le 105° \\ \text{front right} & \text{if } -75° < \theta \le -15° \\ \text{right} & \text{if } -105° < \theta \le -75° \\ \text{back left} & \text{if } 105° < \theta \le 165° \\ \text{back right} & \text{if } -165° < \theta \le -105° \\ \text{back} & \text{else.} \end{cases}$$

(8)

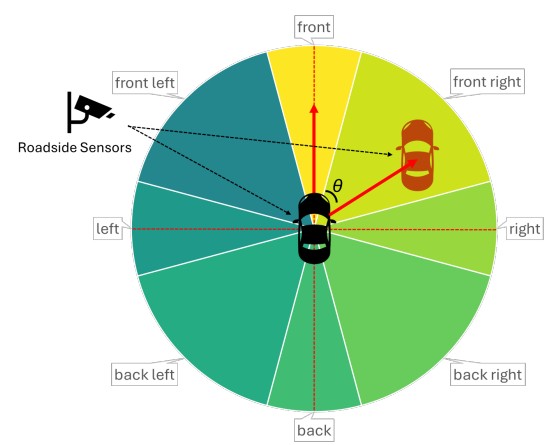

Figure 9: Illustration of the eight spatial regions used to categorize the relative positions of objects in traffic scenes. In this example, the orange car is located to the front right of the black car.

We focus on objects that remain in motion throughout the video. The motion direction of each object is computed based on the difference between its 3D coordinates in consecutive frames. To determine the relative position between two objects, we measure the angle $\theta$ between the motion direction of the moving object and the vector connecting it to another object. Subsequently, the relative position of the second object with respect to the moving object is classified according to the angular criteria defined in Eq. 8. We then divide the spatial relationship into eight distinct regions: front, front left, left, front right, right, back left, back right, and back. Figure 9 illustrates the angular division used to classify the relative position of objects in our TUMTraf VideoQA dataset.

## B. Benchmark Analysis

### B.1. Impact of Frame Number on TraffiX-Qwen Performance

Table 7: Impact of the number of frames on the performance of the TraffiX-Qwen baseline model on the validation set. We report results using spatial pooling as the sampling strategy.

| Models | Frames | Object Captioning | | | Spatio-Temporal Grounding | | | Multi-Choice QA | | | | | |
| | | BLEU_4 | METEOR | SPICE | Temp. Eq↓ | Spa Eq↓ | SP Eq↓ | Positioning | Counting | Motion | Class | Existence | Overall |
|---|---|---|---|---|---|---|---|---|---|---|---|---|---|
| TraffiX-Qwen-0.5B | without | 31.78 | 32.94 | 39.72 | 0.1332 | 0.1979 | 0.2739 | 70.18 | 59.98 | 80.45 | 56.74 | 70.62 | 67.59 |
| | 1 | 33.31 | 34.10 | 42.89 | 0.1205 | 0.1913 | 0.2601 | 71.15 | 76.92 | 82.56 | 67.13 | 84.01 | 76.35 |
| | 11 | 34.58 | 34.85 | 45.12 | 0.1220 | 0.1888 | 0.2594 | 71.92 | 76.89 | 82.94 | 70.54 | 88.84 | 78.97 |
| | 101 | 34.99 | 35.24 | 46.35 | 0.1220 | 0.1892 | 0.2600 | 72.00 | 80.56 | 82.59 | 69.84 | 89.21 | 78.84 |
| | Diff. | +3.21 | +2.30 | +6.63 | -0.0112 | -0.0087 | -0.0139 | +1.82 | +20.58 | +2.14 | +13.10 | +18.59 | +11.25 |
| TraffiX-Qwen-7B | without | 31.80 | 34.66 | 40.74 | 0.1332 | 0.1905 | 0.2710 | 73.32 | 63.93 | 81.44 | 58.65 | 77.72 | 71.01 |
| | 1 | 33.06 | 35.16 | 44.33 | 0.1094 | 0.1791 | 0.2418 | 76.50 | 78.23 | 83.36 | 69.12 | 84.78 | 78.20 |
| | 11 | 35.38 | 36.53 | 47.40 | 0.1078 | 0.1759 | 0.2395 | 76.50 | 81.05 | 82.40 | 72.93 | 87.09 | 80.93 |
| | 101 | 36.74 | 36.42 | 47.32 | 0.1083 | 0.1737 | 0.2382 | 76.56 | 81.94 | 84.33 | 73.42 | 89.44 | 81.14 |
| | Diff. | +4.94 | +1.76 | +6.58 | -0.0249 | -0.0168 | -0.0328 | +3.24 | +18.01 | +2.89 | +14.77 | +11.72 | +10.13 |

To assess the extent to which the baseline model learns from visual tokens and how much it attempts to fabricate answers, we conduct a series of ablation studies. We investigate the impact of the number of frames on TUMTraf VideoQA performance, as detailed in Table 7. Additionally, we include an extreme case where no visual information is provided to the model, and the train baseline model was prompted to answer questions directly.

The experimental results reveal intriguing phenomena in both the 0.5B and 7B models. First, when no visual input is provided, and the model relies solely on the question to generate answers, the baseline model could still reach relatively high performance across all three tasks. This demonstrates the model's inherent reasoning capabilities are probably derived from the question alone and highlights that, in domain-specific datasets such as traffic scenarios, the model appears to learn

and exploit underlying text-based patterns and biases present in the data, which may contribute to its ability to fabricate seemingly accurate responses without actual visual grounding.

Besides, introducing visual input is found to be crucial for correctly solving TUMTraf VideoQA tasks. Across all three tasks, the results consistently show that increasing the number of input frames will improve model performance. Notably, the improvements are most pronounced when moving from no visual input to 1 frame and from 1 frame to 11 frames. However, the performance gains became less significant when increasing the input from 11 frames to 101 frames. This diminishing improvement may be attributed to the inherent difficulty of LLMs in effectively extracting visual context from a large number of tokens. For the 0.5B baseline model, the performance with 11 frames is nearly equivalent to that with 101 frames, reflecting its relatively limited in-context learning capabilities. Therefore, effectively representing video data and addressing the hallucination problem of VLMs in such domain-specific scenarios are critical directions for future research.

Furthermore, the increase in the number of frames has varying impacts on different task types, with substantial differences observed. This variation also indirectly reflects how much the model learns from visual input and how much it affects the reasoning process. For Multi-Choice QA tasks, the gains for positioning and motion categories are the smallest, ranging from only 1.82% to 3.24%. It indicates that the model still struggles to extract answers from visual information effectively based on the current model architecture. In contrast, for counting, class, and existence tasks, the performance improvements exceed 10%, which suggests that VLMs effectively extract features and answer questions in these cases.

### B.2. Performance Evaluation under Consistent Frame Settings

Table 8: Performance of various models under a unified 101-frame input setting. Qwen2-VL uses 96 due to architectural constraints. We report results across three sub-tasks: object captioning, spatio-temporal grounding, and multi-choice QA.

| Model | Size | Object Captioning | | | Spatio-Temporal Grounding | | | Multi-Choice QA | | | | | Overall |
|---|---|---|---|---|---|---|---|---|---|---|---|---|---|
| | | BLEU_4 | ROUGE_L | CIDEr | Temp.E↓ | Spa.E↓ | ST.E↓ | Positioning | Counting | Motion | Class | Existence | |
| LLaVA-OneVision | 0.5B | 0.0 | 0.9 | 0.0 | 1.00 | 1.00 | 1.00 | 25.6 | 25.1 | 11.9 | 12.4 | 0.6 | 15.1 |
| | 7B | 0.0 | 0.5 | 0.0 | 1.00 | 1.00 | 1.00 | 26.4 | 24.1 | 13.6 | 13.0 | 0.8 | 15.6 |
| Qwen2-VL | 2B | 3.8 | 15.1 | 0.18 | 0.65 | 0.68 | 0.73 | 31.2 | 38.4 | 58.2 | 55.4 | 74.5 | 51.5 |
| | 7B | 5.0 | 14.9 | 0.14 | 0.70 | 0.70 | 0.76 | 31.8 | 55.3 | 54.3 | 50.0 | 75.8 | 53.4 |
| Video-LLaMA2 | 7B | 0.0 | 7.1 | 0.0 | 1.00 | 1.00 | 1.00 | 27.6 | 27.5 | 26.3 | 18.7 | 52.2 | 30.4 |
| TraffiX-Qwen | 0.5B | 35.0 | 50.4 | 2.52 | 0.12 | 0.19 | 0.26 | 72.0 | 80.6 | 82.6 | 69.8 | 89.2 | 78.8 |
| | 7B | 36.7 | 52.0 | 2.56 | 0.11 | 0.17 | 0.24 | 76.6 | 81.9 | 84.3 | 73.4 | 89.4 | 81.1 |

To further clarify whether TraffiX-Qwen's performance advantage is due to more frame input, we conduct additional experiments using a unified 101-frame setting for all evaluated models. As open-source video LLMs models often adopt model-specific frame sampling strategies that are tightly coupled with architecture and training paradigms, many existing models (e.g., LLaVA-OneVision, Video-LLaMA2) are not specifically designed to handle long video sequences.

As shown in Table 8, the results reveal several key observations. LLaVA-OneVision and Video-LLaMA2 exhibit notably poor performance when evaluated with 101-frame inputs, indicating their limited capability in modeling long-range temporal dependencies. In contrast, Qwen2-VL demonstrates improved performance with increased frame input, suggesting that its architecture can leverage additional temporal information to enhance reasoning. Most notably, TraffiX-Qwen consistently achieves superior results across all metrics, even under the same input frame conditions. These results show that TraffiX-Qwen achieves robust fine-grained video understanding not merely through longer inputs, but through effective temporal reasoning and structured object representation.

### B.3. Visualization of Multi-Choice QA Results

Figure 10a presents a radar chart depicting the performance of open-source models on the Multi-Choice QA task. The results indicate substantial variability in zero-shot performance across different question types, with each model exhibiting strengths in specific categories. Notably, tasks requiring positioning skills, such as 3D scene understanding, pose significant challenges for all models, suggesting that this question type demands advanced spatial reasoning capabilities, which remain a limitation for current LLMs.

Figure 10b illustrates the performance of TUMTraf VideoQA fine-tuned baseline models. Fine-tuning leads to a notable improvement in overall performance, particularly for the 7B parameter model, which consistently outperforms the lightweight

0.5B model across multiple dimensions. However, the performance gap is not overwhelmingly large, indicating that lightweight models retain considerable practical value and can effectively handle the majority of tasks.

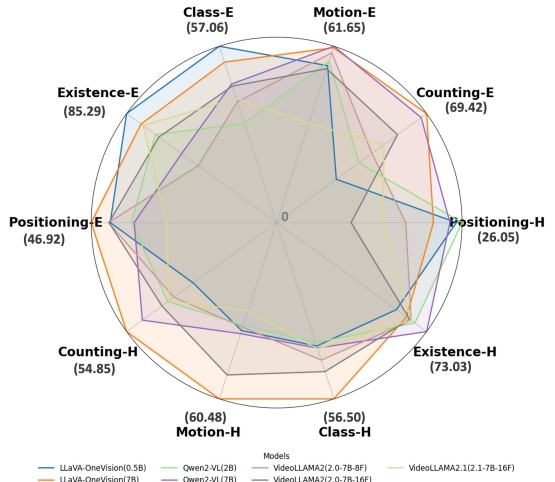 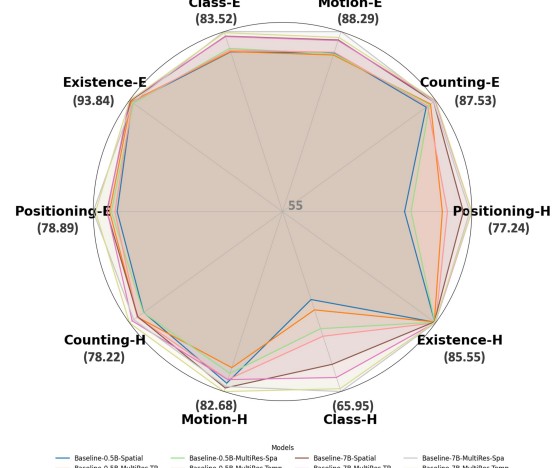

(a) Performance radar chart of the open-source models on the TUMTraf VideoQA Multi-Choice QA task.

(b) Performance radar chart of the TraffiX-Qwen baseline on the TUMTraf VideoQA Multi-Choice QA task.

Figure 10: Results visualization for the open-source models and TraffiX-Qwen baseline models on the Multi-Choice QA.

### B.4. Example of MultiRes Token Pruning

We present several examples of multi-resolution similarity-based token pruning techniques applied to video data from our dataset. As shown in Figure 11, while this approach maintains high resolution to a certain extent, its lack of semantic-aware selection capabilities may result in the loss of task-critical information in certain scenarios. Specifically, it mainly preserves visual tokens for moving vehicles and dynamic objects, such as swaying trees, while pruning stationary vehicles as background information due to their lack of motion. It shows its effectiveness in separating dynamic objects from static backgrounds but also highlights the need for improvement in handling the rest of the important traffic participants.

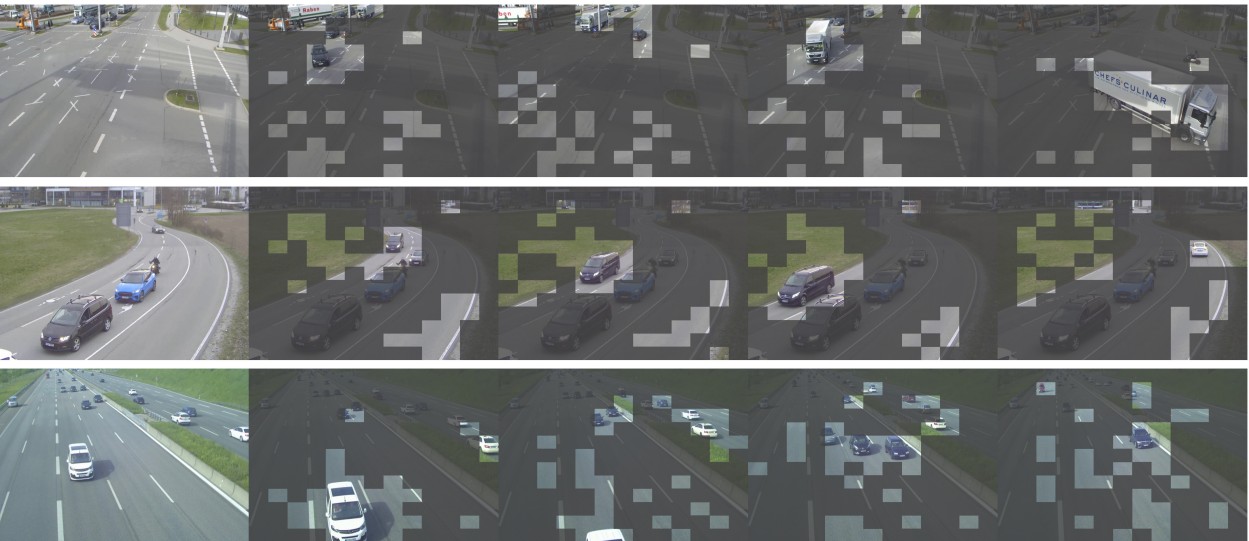

Figure 11: Illustration of cosine similarity-based token pruning, with dark-colored patches representing discarded tokens and preserved ones highlighted. We demonstrate the three samples on highways, country roads, and intersections separately.

### B.5. System Prompt

We craft a dedicated system prompt for our experiments with the TUMTraf VideoQA dataset. Figure 12 presents the prompt used in the experiments. The prompt is adopted across both open-source models and fine-tuned TraffiX-Qwen baseline to ensure fair and consistent evaluation across different models.

> **System Prompt:**
> You are an AI assistant specializing in the analysis of traffic scenes from surveillance footage. Each object's position at a specific moment in the video is represented as a tuple: (c, nf, x, y), where c is the unique identifier for the object, nf is the normalized timestamp of the frame (a float between 0 and 1), and x and y are the normalized coordinates (also between 0 and 1) of the object's position within that frame. Provide precise and informed responses to the questions.

Figure 12: The system prompt used in the experiments of TUMTraffic-VideoQA dataset.

### B.6. Qualitative Evaluations of Spatio-Temporal Object Grounding

Figures 13 through 16 illustrate several qualitative examples of spatio-temporal object grounding, highlighting the challenges and limitations of the task. Figure 13 presents an example where the referred object is a fire truck parked at the roadside, visible throughout the entire video from start to finish. The baseline 0.5B model demonstrates satisfactory temporal localization but exhibits some inaccuracies in spatial localization. In contrast, the baseline 7B model achieves more accurate spatial localization but only identifies the temporal range from 0.2s to 2.95s.

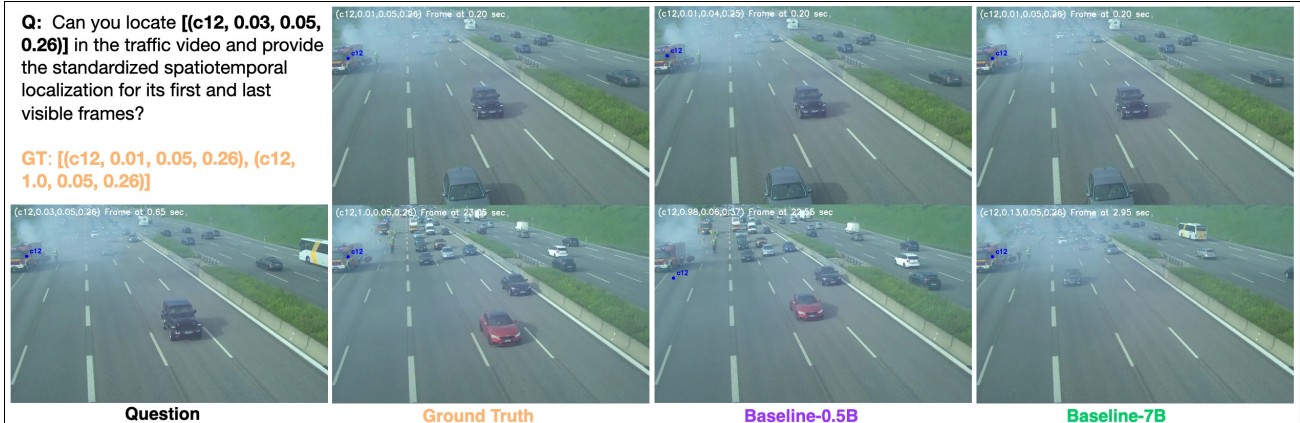

Figure 13: Spatio-Temporal Object Grounding: A fire truck parked at the roadside.

Figure 14 depicts a white car moving along a country road, appearing in the video from 10.10s until the end. The baseline model predictions indicate that the 0.5B model provides a relatively accurate estimate of the initial position, whereas the 7B model exhibits a greater deviation in its ending location.

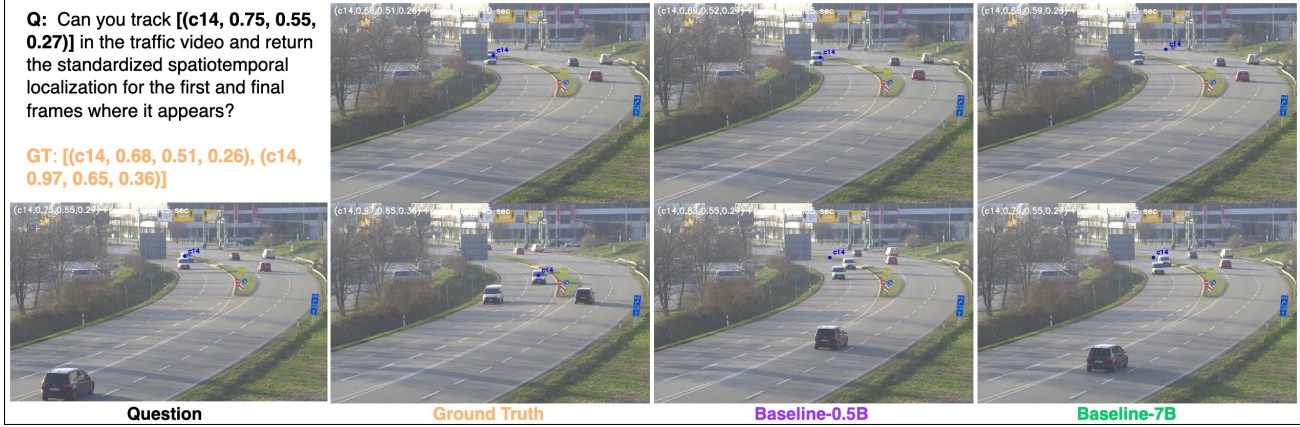

Figure 14: Spatio-Temporal Object Grounding: A white car moving along a country road.

Figure 15 presents the grounding result of a white sedan in a nighttime scene. Due to the object's considerable distance in the reference frame, it appears quite small and makes feature extraction more challenging. Additionally, due to its extended temporal span, the model struggles with cross-frame object association. As a result, both the 0.5B and 7B models fail to accurately capture its end position, instead predicting minimal spatial displacement. This highlights the difficulty of grounding objects with large temporal windows, where precise localization over time remains a significant challenge.

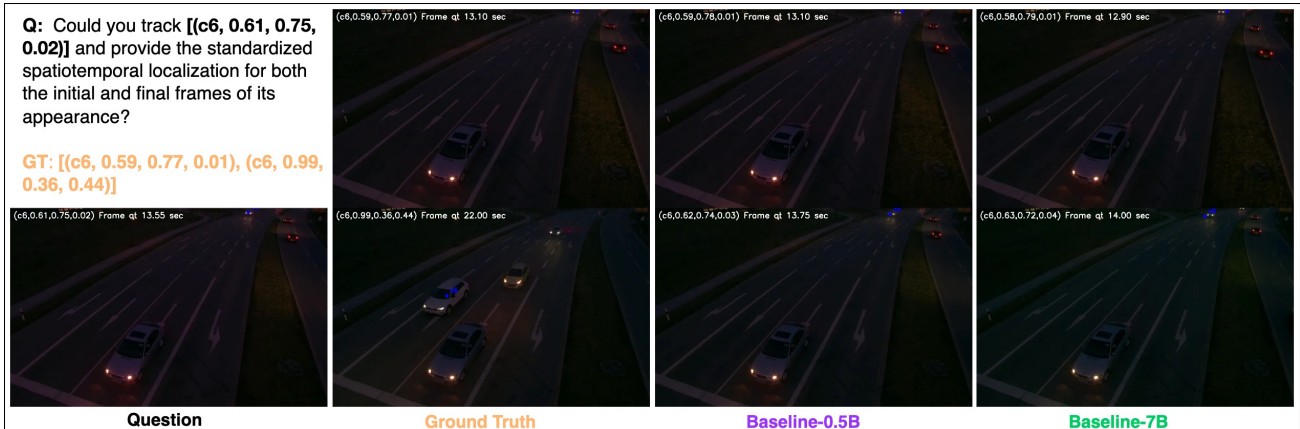

Figure 15: Spatio-Temporal Object Grounding: A white sedan in a nighttime scene.

In Figure 16, we show an example of temporal grounding for a motorcycle at the intersection. Compared to big cars, the grounding of vulnerable traffic participants is much more challenging. Both the 0.5B and 7B baseline models fail to effectively localize the motorcycle in either the temporal or spatial domain, highlighting the difficulty of the task for smaller and less distinct objects.

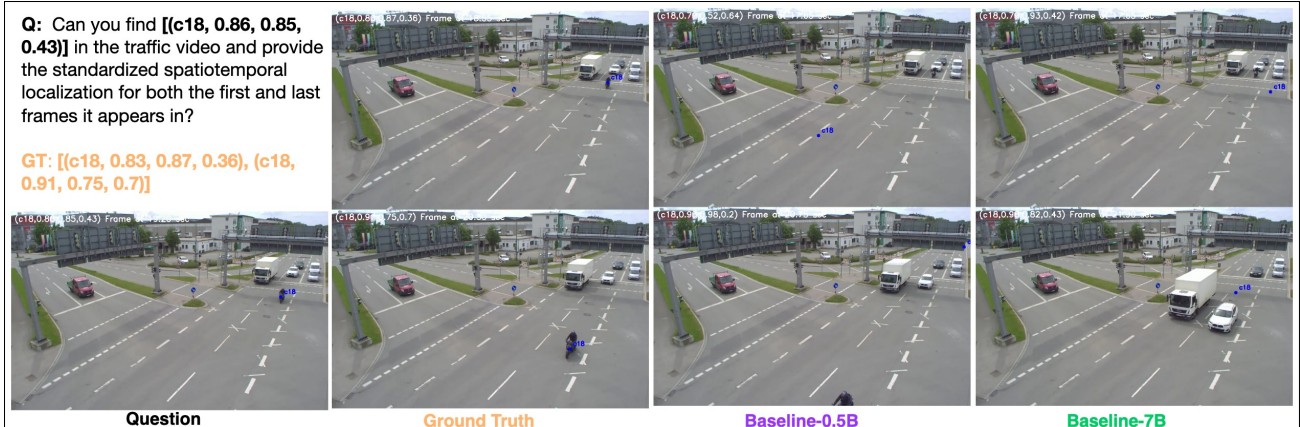

Figure 16: Spatio-Temporal Object Grounding: A motorcycle moving through an intersection.

### B.7. Qualitative Evaluations of Referred Object Captioning

In this section, we present several examples from the referred object captioning task. The left side of each image shows the object to be described, while the right side includes the task description, the corresponding ground truth, and the responses generated by the 0.5B and 7B Traffix-Qwen baseline models. We prompt the model with the question using a list of two tuples that indicate its Spatio-temporal position at two specified timestamps. The experimental results, evaluated using multiple NLG metrics, reveal that the 7B model achieves higher accuracy in describing the appearance details of target objects. However, despite its smaller parameter size, the 0.5B baseline model is also capable of generating satisfactory descriptions, demonstrating its potential practicality in resource-constrained scenarios.

Figure 17 presents a sample to describe an occluded white van. Both the 0.5B and 7B models from the TraffiX-Qwen baseline accurately identify the vehicle as a boxy-shaped white van. However, the 0.5B model introduces extra hallucinations and incorrectly describes the van as having a Volkswagen logo, which is not present in the image. Both the 0.5B and 7B models achieve relatively high metric scores, with the 7B model performing better, particularly in BLEU-4 and SPICE.

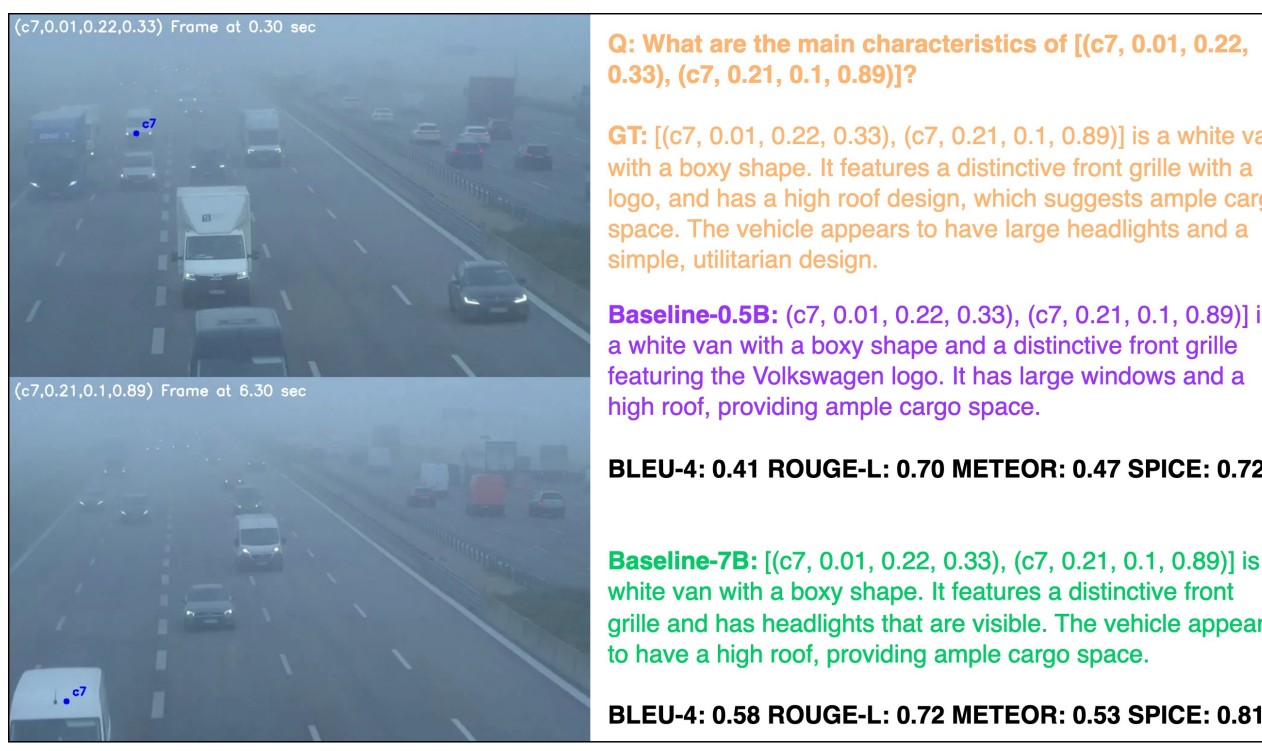

Figure 17: Referred Object Captioning Example: A partially occluded white van with a boxy shape.

Figure 18 illustrates a scenario to describe a dark-colored sedan based on two perspectives captured at different timestamps in the video. The ground truth description from ChatGPT-4o accurately specifies the color as dark purple, while the TraffiX-Qwen baseline, with both the 0.5B and 7B version, classify the vehicle color as black, a visually similar designation. Regarding vehicle type, the 0.5B model identifies it as a hatchback, whereas the 7B model recognizes it as an SUV. Moreover, the 7B model detects distinctive alloy wheels, aligning with the description in ground truth. The quantitative evaluation across four metrics indicates that the 7B model slightly outperforms the 0.5B model, with the most significant improvement observed in the SPICE metric.

Figure 19 presents a case where the question refers to a bus with a distinctive green roof. In the Traffix-Qwen baseline, the 0.5B model incorrectly describes it as a white van with a boxy shape, whereas the 7B model accurately identifies it as a bus with green and white colors and provides a corresponding detailed description. It shows that the 7B model achieves better performance than the 0.5B model for this sample. However, in terms of NLG metrics, both descriptions receive the same ROUGE-L score, which is not a reasonable reflection of their accuracy differences. Among the four reported metrics, SPICE captures the quality of descriptions more effectively. To address such limitations, some studies have introduced LLMs-based evaluation metrics for assessing model performance, which will be explored as part of our future work.

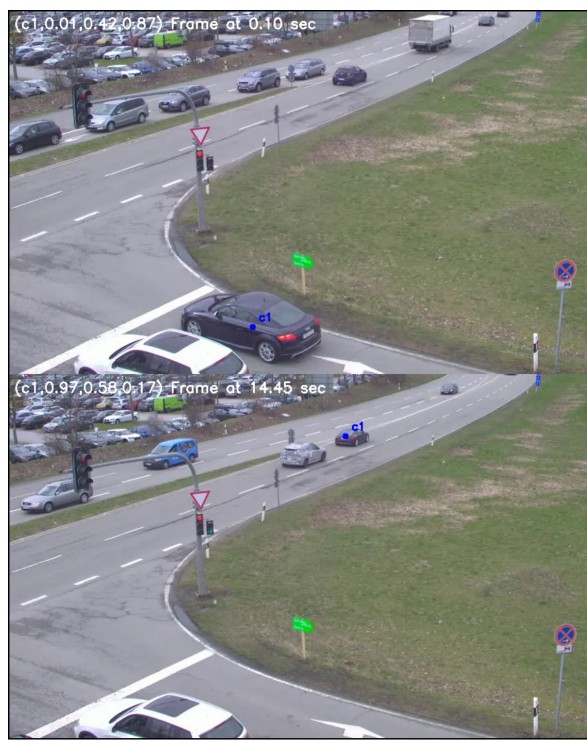

**Q: What are the main characteristics of [(c1, 0.01, 0.42, 0.87), (c1, 0.97, 0.58, 0.17)]?**

**GT:** [(c1, 0.01, 0.42, 0.87), (c1, 0.97, 0.58, 0.17)] is a dark purple coupe with a sleek design and distinctive rounded roofline. It features sporty alloy wheels and a modern rear light design.

**Baseline-0.5B:** [(c1, 0.01, 0.42, 0.87), (c1, 0.97, 0.58, 0.17)] is a black hatchback vehicle with a sleek design. It features distinctive rear lights and a modern tailgate. The vehicle has a compact size, making it suitable for urban environments.

**BLEU-4: 0.33 ROUGE-L: 0.61 METEOR: 0.38 SPICE: 0.38**

**Baseline-7B:** [(c1, 0.01, 0.42, 0.87), (c1, 0.97, 0.58, 0.17)] is a black SUV with a sleek design. It features distinctive alloy wheels and has a modern aesthetic, characterized by smooth lines and a spacious appearance.

**BLEU-4: 0.39 ROUGE-L: 0.69 METEOR: 0.42 SPICE: 0.47**

Figure 18: Referred Object Captioning Example: A dark-purple-colored sedan from two perspectives.

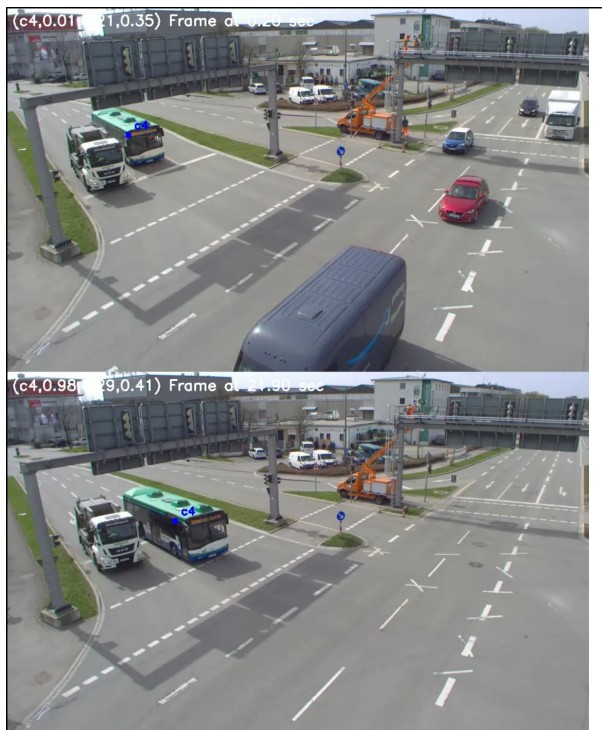

**Q: What are the main features of [(c4, 0.01, 0.21, 0.35), (c4, 0.98, 0.29, 0.41)] captured in the traffic footage?**

**GT:** [(c4, 0.01, 0.21, 0.35), (c4, 0.98, 0.29, 0.41)] is a green bus with a distinctive design featuring multiple roof vents. It has a streamlined shape typical of public transport vehicles.

**Baseline-0.5B:** [(c4, 0.01, 0.21, 0.35), (c4, 0.98, 0.29, 0.41)] is a white van with a boxy shape and a distinctive front grille featuring the Volkswagen logo. It has large windows and a high roof, providing ample cargo space.

**BLEU-4: 0.29 ROUGE-L: 0.58 METEOR: 0.32 SPICE: 0.34**

**Baseline-7B:** [(c4, 0.01, 0.21, 0.35), (c4, 0.98, 0.29, 0.41)] is a bus featuring a green and white color scheme with blue accents. It has a distinctive design with a prominent green roof and blue accents along the lower part of the vehicle.

**BLEU-4: 0.30 ROUGE-L: 0.58 METEOR: 0.34 SPICE: 0.48**

Figure 19: Referred Object Captioning Example: A bus with a distinctive green roof.

# C. Dataset Examples

## C.1. Sample Videos

The TUMTraf VideoQA dataset encompasses a diverse and highly engaging collection of traffic scenarios, capturing a wide range of complex real-world traffic situations and weather conditions. These scenarios cover various traffic dynamics and environmental factors, making the dataset suitable for evaluating models across different conditions. We showcase several representative scene types to illustrate the diversity and characteristics of our dataset more intuitively.

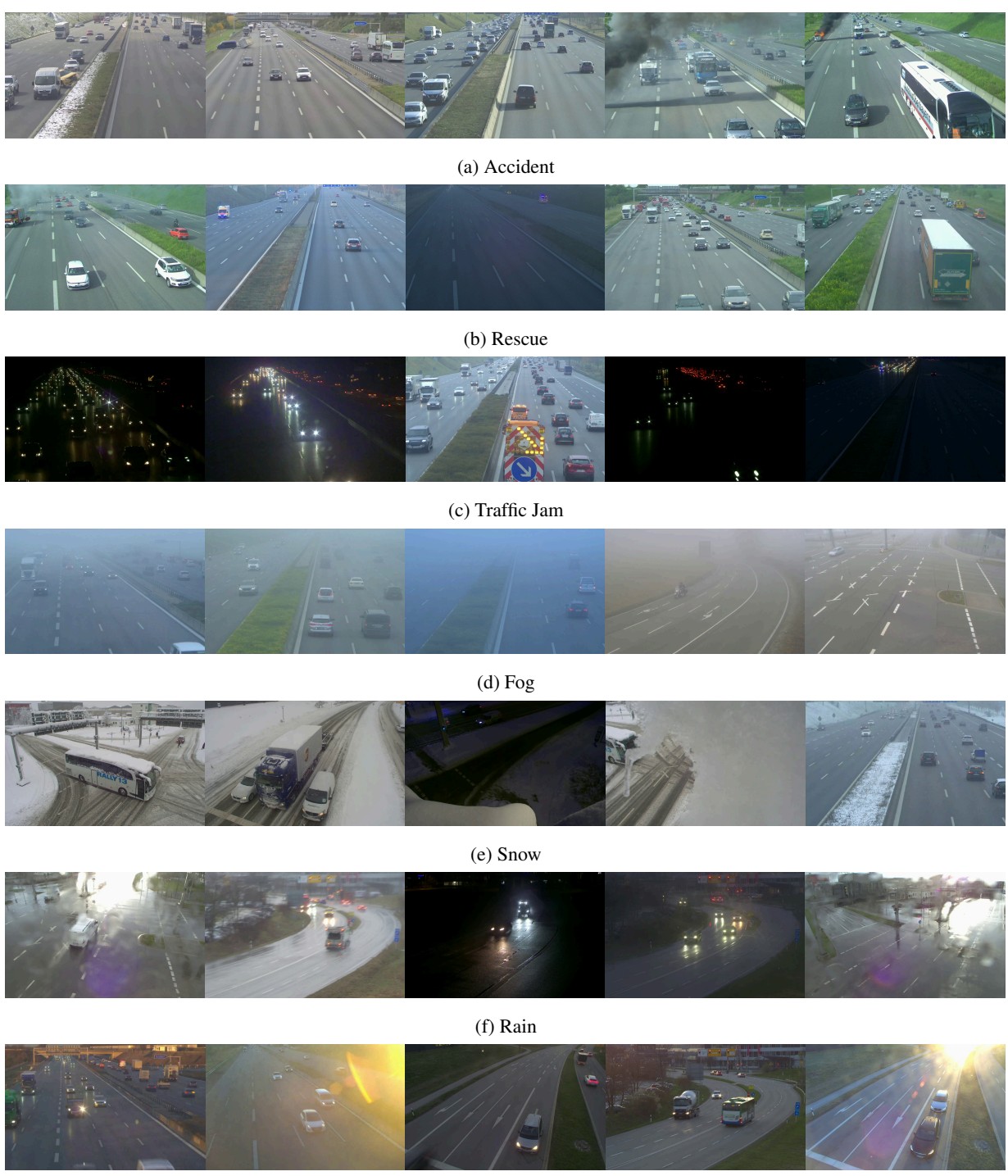

(a) Accident

(b) Rescue

(c) Traffic Jam

(d) Fog

(e) Snow

(f) Rain

(g) Dawn & Dusk

The depicted scenarios include but are not limited to: Traffic Accidents 20a, demonstrating various types and severities of collisions; Rescue Operations 20b, capturing emergency vehicle actions under special circumstances; Traffic Jams 20c, reflecting congestion during peak hours or unexpected events; and scenes under diverse weather conditions, such as Fog 20d, Snow 20e, and Rain 20f, showcasing the dataset's adaptability to complex environments. Additionally, the dataset includes scenarios with unique lighting conditions, such as Dawn and Dusk 20g, simulating traffic dynamics in low-light settings.

## C.2. Question Templates

In this section, we provide some representative examples of question templates for each task. Figures 21 through 25 show templates for the five categories in the Multi-Choice QA task. Figure 26 provides templates for the Spatio-Temporal Object Grounding task and Figure 27 presents templates for the Referred Object Captioning task.

---

Question Template Examples for Positioning-Easy:
Q: "What do you see to the {relative_position} of {object_id} at {normalized_frame} of the video duration?"
Q: "What is present to the {relative_position} side of {object_id} at {normalized_frame} of the video duration?"
Q: "What exists to the {relative_position} of {object_id} at {normalized_frame} of the video duration?"
Q: "What can be observed to the {relative_position} of {object_id} at {normalized_frame} of the video duration?"
Q: "What is on the {relative_position} side relative to {object_id} at {normalized_frame} of the video duration?"
Q: "What can be found to the {relative_position} of {object_id} at {normalized_frame} of the video duration?"
Q: "What can you observe to the {relative_position} side of {object_id} at {normalized_frame} of the video duration?"
Q: "At {normalized_frame} of the video duration, what is visible to the {relative_position} of {object_id}?"
Q: "At {normalized_frame} of the video duration, what can be seen to the {relative_position} of {object_id}?"

Question Template Examples for Positioning-Hard:
Q: "How is {object_id_2} positioned with respect to {object_id_1} at {normalized_frame} of the video duration?"
Q: "Can you specify the location of {object_id_2} relative to {object_id_1} at {normalized_frame} of the video duration?"
Q: "What is the relative location of {object_id_2} to {object_id_1} at {normalized_frame} of the video duration?"
Q: "Can you describe where {object_id_2} is in relation to {object_id_1} at {normalized_frame} of the video duration?"
Q: "Where is {object_id_2} located relative to {object_id_1} at {normalized_frame} of the video duration?"
Q: "Can you describe the relative position of {object_id_2} to {object_id_1} at {normalized_frame} of the video duration?"
Q: "Where can {object_id_2} be found relative to {object_id_1} at {normalized_frame} of the video duration?"
Q: "Can you tell the relative location of {object_id_2} compared to {object_id_1} at {normalized_frame} of the video duration?"

---

Figure 21: Example Positioning question templates. {object_id}, {object_id_1}, and {object_id_2} represent the objects being inquired about, {normalized_frame} is a placeholder for a specific moment in the video duration, and {relative_position} represents the relative position.

```
Question Template Examples for Counting-Easy:
```
Q: "How many {class_name_pl} are captured in the video?"
Q: "How many {class_name_pl} are visible in the traffic footage?"
Q: "How many {class_name_pl} can you detect in the video?"
Q: "How many {class_name_pl} are observable in the video?"
Q: "How many {class_name_pl} can be seen in the video?"
Q: "How many instances of {class_name_pl} are there in the video?"
Q: "How many {class_name_pl} does the video show?"

```
Question Template Examples for Counting-Hard:
```
Q: "Can you identify how many {class_name_pl} are always {motion_status} in the video?"
Q: "In the traffic footage, how many {class_name_pl} are {motion_status} for the whole duration?"
Q: "How many {class_name_pl} are present to the {relative_position} side of {object_id} at {normalized_frame} of the video duration?"
Q: "At {normalized_frame} of the video duration, how many {class_name_pl} exist to the {relative_position} of {object_id}?"
Q: "How many {class_name_pl} are visible to the {relative_position} side of {object_id} at {normalized_frame} of the video duration?"

Figure 22: Example Counting question templates. {class_name_pl} is a placeholder for the plural form of the object class being inquired about, {object_id} is a placeholder for the representation of the object being inquired about, {normalized_frame} is a placeholder for a specific moment in the video duration, {relative_position} represents the relative position, and {motion_status} is a placeholder for the motion status.

```
Question Template Examples for Motion-Easy:
```
Q: "What is the moving status of {object_id}?"
Q: "Can you report the motion status of {object_id}?"
Q: "What's the movement state of {object_id}?"
Q: "What's the movement status of {object_id}?"
Q: "What's the activity status of {object_id}?"
Q: "How would you describe the motion status of {object_id}?"
Q: "What is your description of {object_id}'s motion status?"
Q: "How would you define the movement status of {object_id}?"
Q: "Can you outline the motion status of {object_id}?"

```
Question Template Examples for Motion-Hard:
```
Q: "Is the motion status of {object_id_1} equal to that of {object_id_2}?"
Q: "Are {object_id_1} and {object_id_2} in the same motion state?"
Q: "Is {object_id_1}'s motion status equivalent to {object_id_2}'s?"
Q: "Are the motion statuses of {object_id_1} and {object_id_2} the same?"
Q: "Are the motion states of {object_id_1} and {object_id_2} the same?"
Q: "Do {object_id_1} and {object_id_2} have matching motion statuses?"
Q: "Is {object_id_1}'s motion status identical to {object_id_2}'s?"
Q: "Do {object_id_1} and {object_id_2} share the same motion status?"

Figure 23: Example Motion question templates. {object_id}, {object_id_1}, and {object_id_2} represent the objects being inquired about.

```
Question Template Examples for Class-Easy:
```
Q: "Which class does {object_id} belong to?"
Q: "What category is {object_id} classified under?"
Q: "What is the classification of {object_id}?"
Q: "How is {object_id} categorized?"
Q: "What class label can be given to {object_id}?"
Q: "What is the specific class of {object_id}?"
Q: "How is {object_id} classified?"
Q: "What is the category classification of {object_id}?"
Q: "What type of object is {object_id}?"

```
Question Template Examples for Class-Hard:
```
Q: "Are the {object_id_1} and {object_id_2} of the same type?"
Q: "Is the class of {object_id_1} the same as the class of {object_id_2}?"
Q: "Do {object_id_1} and {object_id_2} belong to the same category?"
Q: "Are {object_id_1} and {object_id_2} from the same category?"
Q: "Are the classes of {object_id_1} and {object_id_2} identical?"
Q: "Do the classes of {object_id_1} and {object_id_2} match?"
Q: "Does {object_id_1} belong to the same category as {object_id_2}?"
Q: "Do {object_id_1} and {object_id_2} share the same class?"
Q: "Is {object_id_1} in the same class category as {object_id_2}?"

Figure 24: Example Class question templates. {object_id}, {object_id_1}, and {object_id_2} represent the objects being inquired about.

```
Question Template Examples for Existence-Easy:
```
Q: "Are there any {class_name_pl} visible in the video?"
Q: "Are any {class_name_pl} present in the traffic footage?"
Q: "Do you spot any {class_name_pl} in the video?"
Q: "Are there any instances of {class_name_pl} visible in the traffic footage?"
Q: "Do any {class_name_pl} make an appearance in the video?"
Q: "Can you spot any {class_name_pl} within the traffic video?"
Q: "Can you confirm the presence of {class_name_pl} in the video?"

```
Question Template Examples for Existence-Hard:
```
Q: "Are there any {class_name_pl} that are {motion_status} for the whole video?"
Q: "Do any {class_name_pl} appear to be {motion_status} for the whole duration of the traffic footage?"
Q: "Can you spot any {class_name_pl} to the {relative_position} of {object_id} at {normalized_frame} of the video duration?"
Q: "Can any {class_name_pl} be seen to the {relative_position} of {object_id} at {normalized_frame} of the video duration?"
Q: "Are there {class_name_pl} visible to the {relative_position} side of {object_id} at {normalized_frame} of the video duration?"

Figure 25: Example Existence question templates. {class_name_pl} is a placeholder for the plural form of the object class being inquired about, {object_id} is a placeholder for the representation of the object being inquired about, {normalized_frame} is a placeholder for a specific moment in the video duration, {relative_position} represents the relative position, and {motion_status} is a placeholder for the motion status.

---

Question Template Examples for Spatio-Temporal Object Grounding:
Q: "Can you track {object_id} in the traffic video and submit the standardized spatiotemporal localization for the first and final frames where it appears?"
Q: "Can you trace {object_id} in the traffic footage and provide the standardized spatiotemporal localization for both the first and last frames of its presence?"
Q: "Can you find {object_id} in the traffic video and provide the standardized spatiotemporal localization for both the first and last frames it appears in?"
Q: "Can you identify {object_id} in the traffic video and provide the standardized spatiotemporal localization for its first and last visible frames? The output should consist of two tuples formatted as (id, nf, x, y), where id is the object's unique identifier, nf is the normalized frame number of detection, and x and y are the normalized coordinates of the bounding box center in each frame."
Q: "Can you locate {object_id} in the traffic video and provide the standardized spatiotemporal localization for its first and last visible frames? The output should be a list containing two tuples, with each tuple structured as (id, nf, x, y). In this format, id denotes the unique identifier of the object, nf represents the normalized frame number in which the object is detected, and x and y are the normalized coordinates of the object's bounding box center within the respective frame."

---

Figure 26: Example Spatio-Temporal Object Grounding question templates. {object_id} is a placeholder for the representation of the object being inquired about.

---

Question Template Examples for Referred Object Captioning:
Q: "What are the main features of {object_id} captured in the traffic footage?"
Q: "What are the key details of {object_id}?"
Q: "What are the main visual characteristics of {object_id}?"
Q: "What are the main characteristics of {object_id}?"
Q: "What are the distinguishing features of {object_id}?"

---

Figure 27: Example Referred Object Captioning question templates. {object_id} is a placeholder for the representation of the object being inquired about.

