# OpenReview forum: "TUMTraf VideoQA: Dataset and Benchmark for Unified Spatio-Temporal Video Understanding in Traffic Scenes"
_ICML.cc/2025/Conference — ICML 2025 poster_

### Official Review · Reviewer_mL1S · 2025-03-12

**Overall Recommendation:** 3

**Summary:**

The paper introduces TraffiX-VideoQA, a benchmark for evaluating spatio-temporal video understanding in traffic scenes. It provides 1,000 videos, 85,000 QA pairs, 2,300 object descriptions, and 5,700 grounding annotations, covering diverse traffic conditions. The authors propose TraffiX-Qwen, a baseline model leveraging multi-resolution visual token sampling to improve temporal reasoning. Experiments compare LLaVA-OneVision, Qwen2-VL, VideoLLaMA2, showing that TraffiX-Qwen achieves superior performance, especially in multi-choice VideoQA. The paper contributes a new dataset, benchmarks existing models, and introduces an improved VideoQA method for real-world traffic scenarios.

## update after rebuttal
The author addressed most of the issues mentioned in my initial review, so I am maintaining my original positive rating.

**Claims And Evidence:**

The paper makes several strong claims, particularly regarding the effectiveness of TraffiX-Qwen and the value of the TraffiX-VideoQA dataset. The dataset is well-constructed, covering diverse traffic conditions, and the model comparison with existing vision-language methods is comprehensive. However, some claims lack sufficient evidence or require further clarification:
1) TraffiX-Qwen processes more frames per video than competing models, which could naturally give it an advantage. It would be more convincing if additional comparisons were made with the same number of frames across all methods.
2) The paper does not introduce new evaluation metrics, even though it discusses performance differences in detail. If the authors claim methodological novelty in evaluation, this should be clarified.
3) While the paper uses SOTA models (YOLOv8, DETR, ByteTrack), it does not analyze their individual contributions. A proper ablation study would strengthen the claim that these choices are optimal for this dataset.
4) The paper asserts that the dataset captures various real-world conditions, but lacks quantitative statistics (e.g., distribution of intersection types). Providing these details would improve the credibility of this claim.

**Essential References Not Discussed:**

The paper provides a solid literature review, but some essential references are missing. For example, the dataset comparison could be expanded to include related benchmarks from autonomous driving and traffic analysis (e.g., Ego4D, BDD100K, Waymo Open Dataset). While these are not VideoQA datasets, they are relevant for understanding real-world traffic dynamics.

**Experimental Designs Or Analyses:**

The experimental setup is mostly well-structured, and the benchmarking of multiple vision-language models provides useful insights. However, there are some concerns regarding experimental fairness and completeness:
1) TraffiX-Qwen processes more frames per video than other models, which could give it a natural advantage. A comparison where all models use the same number of frames would make the results more conclusive.
2) The impact of the multi-resolution visual token sampling strategy is not explicitly studied. Ablation studies would help clarify these factors.
3) The paper presents numerical comparisons but lacks discussions on where and why models fail. A qualitative analysis of failure cases would provide valuable insights.

**Methods And Evaluation Criteria:**

The proposed methods and evaluation criteria are generally well-suited for the task. TraffiX-Qwen’s multi-resolution visual token sampling is a reasonable approach to improve spatio-temporal reasoning while managing computational cost. The TraffiX-VideoQA dataset is well-structured and relevant for evaluating VideoQA in traffic scenes. However, there are some aspects that need further clarification:
1) TraffiX-Qwen sees more frames than competing models, making it difficult to isolate the method’s true impact. A fairer comparison with a fixed number of frames across all models would strengthen the results.
2) While the dataset covers various traffic scenarios, it lacks quantitative details (e.g., the proportion of different road types or intersection structures). Providing such statistics would enhance its credibility.
3) The paper adopts standard metrics but does not introduce new ones. This is fine, but given the complexity of the task, a discussion on potential evaluation limitations (e.g., how frame count affects results) would be valuable.

**Other Comments Or Suggestions:**

Here are some additional comments and suggestions for improving the paper:
1) Some figures, particularly Fig 6 (dataset distribution visualization), could be clearer. Improving resolution or providing more readable labels would help.
2) The paper does not specify the training hyperparameters, batch size, or computational resources required for TraffiX-Qwen. Adding this information would improve reproducibility.
3) The definition of spatio-temporal grounding could be made clearer, particularly in terms of how annotations were generated and whether human verification was involved.
4) Some important dataset details (e.g., intersection type distributions) that are currently missing might be better suited for the appendix.

**Other Strengths And Weaknesses:**

Beyond the points discussed earlier, there are additional strengths and weaknesses worth mentioning:
Strengths:

● The paper is well-structured and written clearly, making it easy to follow, even for researchers unfamiliar with VideoQA.

● The VideoQA task in traffic scenarios is logically structured, making the dataset suitable for future extensions, such as integrating multimodal sensor data (e.g., LiDAR).

● Given the focus on traffic scenarios, this dataset has strong potential for real-world applications in intelligent transportation systems and automated video monitoring.

Weaknesses:

● The proposed TraffiX-Qwen model, while effective, appears computationally expensive due to the multi-resolution visual token sampling strategy. The paper does not discuss training or inference efficiency, which may affect deployment feasibility.

● The dataset is sourced from fixed camera locations, which may introduce a bias toward certain traffic patterns (e.g., urban over rural areas). A discussion of dataset biases and their impact on generalization would be valuable.

**Questions For Authors:**

1) TraffiX-Qwen processes more frames per video than competing models. Could you provide additional results where all models use the same number of frames? This would help determine whether the performance improvement comes from the model design or simply from having access to more temporal information.
2) The paper uses YOLOv8, DETR, and ByteTrack for object detection and tracking, but does not analyze their individual contributions. Have you conducted ablation studies to evaluate how each of these components affects the final performance?
3) The paper mentions that TraffiX-VideoQA covers diverse traffic environments, but does not provide specific statistics on intersection types, traffic conditions, or vehicle categories. Could you share these statistics to support the claim of dataset diversity?
4) The paper does not discuss the computational cost of TraffiX-Qwen. What are the training time, required GPU resources, and inference speed? This information would help assess the feasibility of deploying the model in real-world settings.

**Relation To Broader Scientific Literature:**

The paper provides a solid discussion of prior work, particularly in the areas of VideoQA datasets (NuScenes-QA, DRAMA, DriveLM) and vision-language models (LLaVA, VideoLLaMA2, Qwen2-VL). This establishes a clear foundation for its contributions.

**Theoretical Claims:**

The paper does not contain theoretical claims or formal proofs, as its contributions focus on dataset construction, model development, and empirical evaluation. The proposed multi-resolution visual token sampling is validated experimentally rather than theoretically. While this is reasonable for the scope of the work, a theoretical discussion on how the sampling strategy impacts spatio-temporal reasoning could strengthen the justification.

---

> ### Author Rebuttal · Authors · 2025-03-31
>
> Dear Reviewer,
>
> We appreciate your valuable feedbacks and address your concerns as follows.
>
> Q1: Clarification on whether TraffiX-Qwen’s performance gain stems from more input frames.
>
> A1: As open-source VideoQA models often adopt model-specific frame sampling strategies, which are tightly coupled with architecture and training, some models (e.g., LLaVA-OneVision, Video-LLaMA2) are not designed to handle longer frame sequences. Hence, we used different frame settings that were in line with their original paper and codebase. We also fully agree that comparisons under a unified frame setting would strengthen the results. Therefore, we conduct extra experiments with 101 frame inputs for the models used in our paper. (Qwen2-VL, whose model structure imposes constraints on input frame numbers, is adapted to 96 frames.)
>
>
> |**Model**|**Size**|**BLEU_4**|**ROUGE_L**|**CIDEr**|**Temp.E↓**|**Spa.E↓**|**ST.E↓**|**Pos.**|**Count.**|**Motion**|**Class**|**Exist.**|**Overall**|
> |-|-|-|-|-|-|-|-|-|-|-|-|-|-|
> |LLAVA-OneVision|0.5B|0.0|0.9|0.0|1.0|1.0|1.0|25.6|25.1|11.9|12.4|0.6|15.1|
> ||7B|0.0|0.5|0.0|1.0|1.0|1.0|26.4|24.1|13.6|13.0|0.8|15.6|
> |Qwen2-VL|2B|3.8|15.1|0.18|0.65|0.68|0.73|31.2|38.4|58.2|55.4|74.5|51.5|
> ||7B|5.0|14.9|0.14|0.70|0.70|0.76|31.8|55.3|54.3|50.0|75.8|53.4|
> |VideoLLAMA2|7B|0.0|7.1|0.0|1.0|1.0|1.0|27.6|27.5|26.3|18.7|52.2|30.4|
> |**TraffiX-Qwen**|0.5B|35.0|50.4|2.52|0.12|0.19|0.26|72.0|80.6|82.6|69.8|89.2|78.8|
> ||7B|36.7|52.0|2.56|0.11|0.17|0.24|76.6|81.9|84.3|73.4|89.4|81.1|
>
> Results show that: 1. LLaVA-OneVision and Video-LLaMA2 perform far worse with 101 frames, likely due to the lack of long-range temporal modeling. 2.	Qwen2-VL shows improved performance with more frame inputs, suggesting that additional temporal information can indeed be beneficial. This also highlights Qwen2-VL’s stronger temporal reasoning capability.
>
> Importantly, even under the same frame input setting, TraffiX-Qwen still consistently outperforms all baselines, confirming that the performance gain stems from multiple aspects, not just access to more temporal information. These results will be added to the revised paper.
>
> Q2: The role of YOLOv8, ByteTrack, etc in the pipeline.
>
> A2: We use 2D detectors and trackers only during the TraffiX-VideoQA dataset construction to generate meta-information aligned with human annotations, improving quality and consistency. They are not used in TraffiX-Qwen’s training or inference. Our baseline is a fully end-to-end VLM that processes raw videos and generates answers directly, without external modules. This design simplifies the pipeline and promotes research toward integrated video understanding in the traffic domain.
>
> Q3: Statistics supporting the claim of dataset diversity.
>
> A3: We add 3 figures illustrating dataset distribution at: https://imgur.com/a/eoZXJEb.
>
> The figures statistically support the dataset’s diversity. We group scenes in our dataset into 3 key types: highways (rural), urban intersections (city), and country roads (rural/urban). They show that the dataset aligns well with real-world traffic distributions. The figures will be added in the revised version.
>
> Q4: Missing information on model efficiency and computational cost.
>
> A4: We provide the training and inference details in the paper:
>
> |**Version**|**#Vision**|**#Projector**|**#LLM**|**Inf Speed/QA**|**#Trainable**|**Train Hour**|
> |-|-|-|-|-|-|-|
> |**0.5B**|397.8M|1.8M|493.8M|~1.6s|495.6M|28h|
> |**7B**|397.8M|17.0M|7612.6M|~3.8s|7629.6M|36h|
>
> Inference time is measured as the average to process a 10s–1min video with autoregressive decoding on 1*A100 GPU (no acceleration). We believe further optimization (e.g., quantization, pruning, distillation) is a promising direction for improving deployment efficiency in traffic monitoring. We will include this table and discussion in the revised version.
>
>
> Response to other suggestions:
>
> 1. We have included an initial discussion regarding how sampling strategy impacts spatiotemporal reasoning in Sec 5. We will expand this with new experimental results and discussions.
> 2. Qualitative results for ST-OG and V-ROC have been added to Appendix B.5, B.6, covering both successful and failure cases. For the MC-QA task, we will provide extra qualitative visualizations and analyses.
> 3. Initial analysis of frame count effects on TraffiX-Qwen is provided in Appendix B.1. We will expand this with an extra discussion of how frame sampling impacts temporal reasoning capabilities.
> 4. We agree that including extra AD benchmarks (e.g., Ego4D etc.) brings valuable context and will add them.
> 5. We will add more details on model complexity, and potential acceleration techniques to support practical deployment.
> 6. We agree that using fixed camera views may introduce distributional bias in traffic patterns and will include a discussion on this in limitation.
> 7. We will improve Fig. 6. refine the definition of spatiotemporal grounding, and emphasize the role of human verification during annotation process.

---

> > ### Comment · Reviewer_mL1S · 2025-04-04
> >
> > Thank you for your response. I will keep my positive rating. Additionally, I recommend including this supplementary information in the revised version.

---

> > > ### Author Response · Authors · 2025-04-09
> > >
> > > Dear Reviewer,
> > >
> > > Thank you very much for your positive rating and valuable feedback. We appreciate the recommendation, and we will include the additional experiments and corresponding details based on your suggestions in the revised version to strengthen the paper further.

---

### Official Review · Reviewer_z4TA · 2025-03-12

**Overall Recommendation:** 3

**Summary:**

The paper presents a comprehensive video-language dataset designed for complex traffic video understanding, named TrafficX-VideoQA. Meanwhile, a benchmark is provided, including multiple-choice video question answering, referred object captioning, and spatiotemporal object grounding tasks. Experimental results demonstrate that TrafficX-VideoQA is complex.

**Claims And Evidence:**

The claims made in the submission are supported by clear and convincing evidence.

**Essential References Not Discussed:**

There is a need to incorporate the recent works[1] for more comprehensive analysis.
[1] Towards Surveillance Video-and-Language Understanding: New Dataset, Baselines, and Challenges. CVPR 2024

**Experimental Designs Or Analyses:**

I have checked the experimental designs and analyses. A comprehensive analysis was conducted for three key tasks in the paper.

**Methods And Evaluation Criteria:**

The proposed methods and evaluation criteria make sense for the problem or application at hand.

**Other Comments Or Suggestions:**

Why only focus on multiple-choice video question answering, referred object captioning, and spatiotemporal object grounding tasks? Wouldn't it be better to cover all the tasks of existing video understanding?

**Other Strengths And Weaknesses:**

The writing and the organization are good. And the proposed dataset and benchmark is valuable.

**Questions For Authors:**

1. Why only focus on multiple-choice video question answering, referred object captioning, and spatiotemporal object grounding tasks? Wouldn't it be better to cover all the tasks of existing video understanding?

2. Compared to the reference [1], what are the strengths of the paper?
[1] Towards Surveillance Video-and-Language Understanding: New Dataset, Baselines, and Challenges. CVPR 2024

**Relation To Broader Scientific Literature:**

The paper facilitates further advancements in traffic video analysis and contributes to the development of next-generation traffic foundation models.

**Theoretical Claims:**

I have checked proofs for theoretical claims. For example, the paper states that "most existing efforts primarily focus on driving scenarios and are typically constrained to individual tasks such as question answering, video grounding, or referred multi-object tracking". From Table 1, the data are supported.

---

> ### Author Rebuttal · Authors · 2025-03-30
>
> Dear Reviewer,
>
> Thank you very much for your positive feedback and valuable reviews. Our responses to your comments are detailed below:
>
> Q1. There is a need to incorporate the recent work [1] for a more comprehensive analysis.
>
> A1. Thank you for pointing out this important related work. We have discussed a detailed comparison with it in our response to Q3 below, and we will also include this reference in our final manuscript to improve the completeness and introduce our work more clearly.
>
> Q2. Why only focus on multiple-choice video QA, referred object captioning, and spatiotemporal object grounding?
>
> A2. Thank you for the insightful suggestion. We fully agree that video understanding covers many valuable tasks beyond our current focus, such as action recognition, event localization, video captioning, and anomaly detection. Following the survey [2], we broadly categorize video understanding tasks based on the granularity of required information into abstract, temporal, and spatiotemporal levels.
>
> In traffic scenarios, high-level events are typically composed of interactions among traffic participants. As such, fine-grained spatiotemporal understanding and reasoning serve as a crucial foundation for interpreting complex traffic events.  Based on this, our benchmark focuses first on spatiotemporal understanding through three core tasks in traffic scene comprehension: multiple-choice video QA, spatiotemporal object grounding, and referred object captioning. We sincerely appreciate your suggestion and agree that using VLMs for higher-level semantic tasks such as traffic event recognition is a promising direction. We will add this to the Future Work section of our paper.
>
> Q3. Compared to [1], what are the strengths of the paper?
>
> A3. Thank you for raising this important question. We have reviewed this work and highlighted the following strengths of our dataset and benchmark compared to the work [1] as follows:
>
>  1. **Data Source and Data Novelty**.
>
> Ref [1] is built upon existing datasets, providing additional textual caption annotations without contributing new video data or organizing raw footage. Moreover, most of the videos used in [1] are sourced from the internet. In contrast, our dataset is collected from real-world traffic scenes, ranging over 2 years using multiple intelligent infrastructure systems installed from over 20 diverse perspectives. From over 1,000 hours of raw video data, we curate our benchmark through a semi-automatic data annotation process. Compared to internet videos, our data more accurately reflects the distribution and characteristics of real-world traffic scenarios, offering higher data novelty and practical value. We include figures to support this point as suggested by Reviewer mL1S, A3: https://imgur.com/a/eoZXJEb
>
> 2. **Inherent Limitation for Fine-Grained Spatiotemporal Tasks**.
>
> Ref [1] focuses on video understanding tasks that require abstract semantic information, such as high-level event descriptions. However, such object reference is not well-suited for fine-grained spatiotemporal tasks in traffic scenes. This limitation arises from several factors: the inherent ambiguity of descriptive natural language expressions, the modality gap between visual and linguistic representations, and the high visual similarity of traffic participants in surveillance footage.
>
> This limitation is also clearly illustrated in Ref [1], Figure 1 (top video). The sentence query states: “A black car drove past the corner, and a gray car followed closely behind it.” However, the video scene contains multiple black cars and multiple gray cars, making the reference inherently ambiguous. Such ambiguity prevents the benchmark from supporting more fine-grained VideoQA tasks, particularly those requiring precise object grounding and temporal reasoning.
>
> 3. **Unified Benchmark with Structured Object Representation**.
>
> Our benchmark unifies three fine-grained tasks, i.e., multi-choice QA, referred object captioning, and spatiotemporal object grounding, via a tuple-based spatiotemporal object expression. This design enables benchmarks on more complex tasks, such as relative spatial reasoning, which are critical in the traffic domain but missing from [1].
>
> 4. **Unified Model for VideoQA Benchmark**.
>
> In [1], each proposed task is evaluated separately using existing models like CoCap or UEDVC. While valuable, these approaches are separate for each task and do not support unified understanding. In contrast, we propose TraffiX-Qwen, a unified VLM-based model trained with multi-task learning across all three tasks. This contributes a solid baseline toward generalizable traffic visual foundation models.
>
> [1] Towards Surveillance Video-and-Language Understanding: New Dataset, Baselines, and Challenges. CVPR 2024
>
> [2] Video Understanding with Large Language Models: A Survey. arXiv:2312.17432, 2024.

---

### Official Review · Reviewer_QbCi · 2025-03-13

**Overall Recommendation:** 5

**Summary:**

This paper proposes a new traffic VQA dataset captured from the roadside. The paper proposes three tasks based on the dataset, including Multi-Choice Question Answering, Video Referred Object Captioning, and Spatio-Temporal Object Grounding. The author further proposes a new, unified method to tackle all three tasks. The established benchmark offers a comprehensive analysis and demonstrates the strong performance of the proposed method.

**Claims And Evidence:**

L55 “they often face significant challenges in scalability, generalization to diverse
traffic conditions, and real-world deployment.” Unclear why?

Although the authors attempt to explain this with the following sentence, it is still unclear why the recent large models can address the mentioned challenges, especially the scalability and generalization to diverse traffic conditions and real-world deployment. In other words, the statements justifying the importance of QA in traffic scenes are very vague and not convincing. Note that I am not against the importance of the topic but feel the statement could be improved.

**Essential References Not Discussed:**

Missing highly relevant discussion of work proposing a description for traffic event recognition [1,2] in Related work - Fine-Grained Video Understanding or Language-Based Object Referring.

[1] Agarwal and Chen, Ordered Atomic Activity for Fine-grained Interactive Traffic Scenario Understanding, ICCV 2023

[2] Kung et al., Action-slot: Visual Action-centric Representations for Multi-label Atomic Activity Recognition in Traffic Scenes, CVPR 2024

**Experimental Designs Or Analyses:**

The experimental design is very thorough and comprehensive.

**Methods And Evaluation Criteria:**

The performance of the method is strong and evaluated fairly.

**Other Comments Or Suggestions:**

- It would be great to have a detailed definition of the difficulty level of questions, i.e., single-hop or multi-hop, in the supplementary material.
- Unclear why there are multiple bars for a class of questions in Figure 4 (a)

**Other Strengths And Weaknesses:**

**strengths:**

- the paper is well-written and easy to follow.

**Weakness:**

- It is unclear what is the difference between the proposed Multi-Choice Question Answering and existing VQA work, besides the camera view.
- Unclear why the authors want to investigate different strategies of token sampling as they all perform very similarly in terms of performance across different tasks.
- Unclear why the authors highlight “traffic anomalies” (L40 in the abstract) and “critical corner cases such as traffic accidents” (L96 in the contributions). This would mislead readers to expect to see benchmark results analyzing the performance under these anomaly scenarios.

**Questions For Authors:**

Please see the comments.

**Relation To Broader Scientific Literature:**

The proposed traffic VQA dataset can offer new opportunities for studying more fine-grained traffic understanding topics and facilitate the development of AV.

**Theoretical Claims:**

Yes, I have checked the correctness of all proofs or equations in the main paper.

---

> ### Author Rebuttal · Authors · 2025-03-31
>
> Dear Reviewer,
>
> Thank you very much for your positive feedback and valuable comments. Below, we provide point-by-point responses to the concerns raised.
>
> Q1: What are the differences between the proposed multi-choice QA and existing VQA work, aside from the camera view?
>
> A1: Thank you for the thoughtful question. Beyond the camera perspective, our work differs from previous VQA works in several key aspects:
>
> 1.	The object reference expression in existing video QA works cannot be directly applied to the domain of spatiotemporal traffic scene understanding. We also argue it in reply to **Reviewer z4TA, A3-2**. This is due to the inherent ambiguity of natural language, the modality gap between visual and linguistic representations, and the need for accurate and unique object referring in traffic scenarios. Purely language-based descriptions often fail to achieve precise cross-modal object association. Hence, our work introduces a structured tuple-based object representation, enabling accurate and interpretable fine-grained analysis of traffic videos.
>
> 2.	Our benchmark is specifically designed for unified intelligent traffic monitoring and scene understanding, which is currently missing in existing works. We aim to bridge this gap by providing a dataset and benchmark tailored to traffic scene analysis needs. Our dataset reflects real-world traffic distributions, and we include additional figures to support this point in response to **Reviewer mL1S, A3**:  https://imgur.com/a/eoZXJEb
>
> 3.	We unify three previously disjoint tasks in this domain, i.e., multiple-choice video QA, referred object captioning, and spatiotemporal object grounding, into a single benchmark. This positions our work beyond traditional VQA, offering a more comprehensive and domain-relevant challenge.
>
> Q2: Why investigate different token sampling strategies when they show similar performance across tasks?
>
> A2: Thank you for raising this question. In traffic scenes, particularly in the context of intelligent traffic monitoring and scene understanding, a key domain-specific characteristic is that the visual background tends to remain relatively static within a video.
>
> For existing VLMs, efficient visual representation is critical to overall performance. Therefore, exploring different visual token sampling strategies that remove redundant information while preserving key content is particularly important in this domain. Our study investigates whether tailored visual token strategies can leverage this static background nature to improve efficiency and performance. This motivation is also consistent with some prior works for roadside visual-centric tasks [3][4], where techniques are applied to suppress background redundancy and enhance downstream task performance.
>
> Q3: Why emphasize “traffic anomalies” in the abstract and contributions without dedicated benchmark analysis?
>
> A3: Our intention was to reflect real-world distribution, where traffic anomalies (e.g., accidents) are rare but critical cases. While our benchmark does not focus solely on anomaly analysis, we still provide high-level annotations indicating whether each video contains anomalies. This supports valuable future research focused on anomaly detection and emphasizes the diversity and realism of our dataset. We will revise the content to clarify this.
>
> Q4: Define the difficulty levels of QA questions (e.g., single-hop vs. multi-hop) in the supplementary material.
>
> A4: Thank you for highlighting this point. We currently include a subset of QA templates in Supplementary Material C.2. To improve clarity, we will introduce an additional subsection that clearly defines and categorizes the difficulty levels of QA questions.
>
> Q5: Clarify why there are multiple bars for a class of questions in Figure 4(a).
>
> A5: In Figure 4(a), we plot the distribution of question word counts for each question task/type. Since questions within the same task type can vary significantly in length and structure, we chose to visualize this distribution to better reflect the linguistic diversity of the dataset, rather than simply reporting the total number of questions per type. We will revise the caption to make this clearer.
>
>
> Q6: L55 - The statement justification for the importance of QA in traffic scenes is unclear.
>
> A6: Thank you for pointing out the unclear phrasing. We will revise this part to emphasize the motivation and importance of introducing LLM/VLM for the intelligent traffic scene understanding domain and the development of AV in our revised version.
>
>
> Q7: Missing highly relevant work on traffic event recognition [1,2] in the Related Work section.
>
> A7: We appreciate your suggestion. We will include the missing references and expand the section to properly acknowledge prior work on traffic event recognition.
>
>
> [3] Zekun Zhang et al., Object Detection With Self-Supervised Scene Adaptation, CVPR 2023.
>
> [4] Shuo Wang et al., The 8th AI City Challenge, CVPR-W 2024.

---

> > ### Comment · Reviewer_QbCi · 2025-04-09
> >
> > I appreciate the authors for their very comprehensive response. All my concerns are addressed well. I encourage the authors to revise the final version accordingly for better readability and clarity. I believe the proposed dataset and study can offer good insights and a platform for future traffic scene understanding and thus I am willing to raise my rating from accept to strong accept.

---

> > > ### Author Response · Authors · 2025-04-09
> > >
> > > Dear Reviewer,
> > >
> > > We sincerely appreciate the valuable feedback you provided and your recognition of our work!  We will revise the final version of the paper based on your suggestions to further improve its readability and clarity.

---

### Official Review · Reviewer_9vbN · 2025-03-14

**Overall Recommendation:** 4

**Summary:**

This paper provides a comprehensive dataset tailored for multiple tasks in traffic scenarios. It includes QA such as predicting the weather, counting objects, providing motion status, spatio-temporal grounding, and more. It consists of 1,000 videos, with 85,000 QA pairs, 2,300 object captioning, and 5,700 object grounding annotations. The benchmark tasks include video QA, object captioning, and grounding. The accompanied baseline model is based on Qwen-2 plus multiple important visual modules such as multi-resolution token sampling and temporal/spatial/token pooling. The paper provides the results from both the baseline model as well as other open-sourced multimodal LLMs.

**Claims And Evidence:**

The paper’s data analysis, model comparisons, and ablation studies show how existing open-sourced models perform relatively poorly with fine-grained spatio-temporal tasks, and the authors’ enhanced baseline with multiple techniques still shows the dataset’s complexity.

**Essential References Not Discussed:**

NA.

**Experimental Designs Or Analyses:**

One problem with this dataset is that the baseline models already score relatively high on the QA task already (e.g., 81.95% for overall QA accuracy in Table 3). Would the authors provide a brief discussion on how this dataset, especially the QA part, will remain challenging for upcoming new multimodal LLMs?

**Methods And Evaluation Criteria:**

Yes. The tasks—multi-choice QA, spatio-temporal grounding, and referred object captioning—directly address real roadside surveillance needs like identifying moving vehicles, counting objects, and localizing them over time. The chosen metrics (QA accuracy, spatio-temporal error, and NLG measures on captions) are standard and align with these tasks.

**Other Comments Or Suggestions:**

NA

**Other Strengths And Weaknesses:**

No assets are provided in the submission. The reviewer recommends the authors to release the codebase as well as the dataset upon acceptance.

**Questions For Authors:**

See Section Experimental Designs Or Analyses.

**Relation To Broader Scientific Literature:**

This dataset is a great comprehensive video dataset that is positioned as a challenging benchmark for advancing research in intelligent transportation systems.

**Theoretical Claims:**

No theoretical claims are provided in this paper.

---

> ### Author Rebuttal · Authors · 2025-03-30
>
> Dear Reviewer,
>
> Thank you very much for your positive feedback and insightful questions. We provide detailed responses to your concerns below.
>
> Q1: Would the authors provide a brief discussion on how this dataset, especially the QA part, will remain challenging for upcoming new multimodal LLMs given the high baseline results?
>
> A1:
> During our literature review, we also observed that some VQA benchmarks in vertical application domains, such as earth observation, autonomous driving, and smart cities, tend to achieve relatively high baseline accuracy (generally > 60\%) after fine-tuning (e.g., EarthVQA[1], nuScenes-QA [2], City-3DQA [3]). Our dataset shows a similar trend, indicating that large VLMs can more easily adapt to constrained, domain-specific tasks. However, we believe that the high baseline accuracy does not indicate that the task is simple or lacking in meaningful challenges. Our benchmark involves multiple complex tasks that remain crucial for developing future multimodal LLMs, especially those for traffic-related applications. Below, we provide a detailed discussion:
>
> 1.	Our benchmark unifies three core tasks, i.e., multi-choice QA, referred object captioning, and spatiotemporal object grounding through tuple-based spatiotemporal object expressions for traffic scene understanding. While the QA task demonstrates relatively high accuracy, the other two tasks still exhibit substantial room for improvement. Importantly, as discussed in the paper, techniques that enhance QA performance, such as visual token strategies, do not consistently benefit the other tasks. This reveals that optimizing for QA alone is insufficient, and highlights the need for holistic model designs that jointly address all tasks.
>
> 2.	Even within the multi-choice QA task, simply scaling model size yields diminishing returns. Although the 7B model achieves higher accuracy, the improvement over smaller models (e.g., 0.5B) is marginal. This suggests that simply increasing LLM size or replacing it with newer variants alone is unlikely to bring large improvement in this domain. Together, these findings emphasize the importance of domain-specific model design and training strategies that consider multiple tasks tailored to traffic-centric understanding.
>
> 3.	In real-world traffic applications, where computational resources are often limited, developing lightweight models becomes a key priority. Rather than focusing solely on improving benchmark scores, we encourage future work to explore trade-offs between performance and efficacy with lightweight LLMs and techniques (i.e., pruning, quantization, and knowledge distillation).  Our benchmark and baseline models provide practical references to guide and evaluate such efforts toward building efficient, real-time multimodal systems.
>
> In summary, we believe our dataset and benchmark remain highly important for the development and evaluation of upcoming multimodal LLMs, particularly those tailored to traffic domains. It presents practical challenges, encourages efficient model development, and supports a holistic evaluation across multiple multimodal tasks.
>
> [1] Junjue Wang,  et al., “EarthVQA: Towards Queryable Earth via Relational Reasoning-Based Remote Sensing Visual Question Answering”
>
> [2] Tianwen Qian, et al., “NuScenes-QA: A Multi-modal Visual Question Answering Benchmark for Autonomous Driving Scenario”
>
> [3] Penglei Sun, et al., “3D Question Answering for City Scene Understanding”

---

> > ### Comment · Reviewer_9vbN · 2025-04-04
> >
> > The reviewer is satisfied with the rebuttal and keeps the positive rating.

---

> > > ### Author Response · Authors · 2025-04-09
> > >
> > > Dear Reviewer,
> > >
> > > Thank you very much for your positive rating and for taking the time to review our rebuttal. We’re glad to hear that our responses addressed your concerns, and we sincerely appreciate your recognition of our work.

---

### Decision · Program_Chairs · 2025-05-01

**Decision:**

Accept (poster)

**Comment:**

This work propses a new dataset for evaluating traffic scenarios with evaluation based on question answering spanning tasks like: multi-choice vqa, referred object captioning, and object grounding/tracking. This work also evaluates open VLMs like LLava-OneVision, VideoLlama2, Qwen2-VL, and adapt Qwen-2 with pre-training (on raw data) and "instruction" tuning on QA pairs from the training set of the proposed dataset.

The proposed dataset presents a challenging benchmark to existing VLMs. The reviewers found dataset contribution and evaluation as important for transportation systems, especially for fine-grained evaluation. Hence, I'm inclined to recommend this work further.